# Approximate Forest Completion and Learning-Augmented Algorithms for Metric Minimum Spanning Trees

Nate Veldt [1]   Thomas Stanley [1]   Benjamin W. Priest [2]   Trevor Steil [2]   Keita Iwabuchi [2]   T.S. Jayram [2]
Geoffrey Sanders [2]

## Abstract

Finding a minimum spanning tree (MST) for $n$ points in an arbitrary metric space is a fundamental primitive for hierarchical clustering and many other ML tasks, but this takes $\Omega(n^2)$ time to even approximate. We introduce a framework for metric MSTs that first (1) finds a forest of trees using practical heuristics, and then (2) finds a small weight set of edges to connect disjoint components in the forest into a spanning tree. We prove that optimally solving step (2) still takes $\Omega(n^2)$ time, but we provide a subquadratic 2.62-approximation algorithm. In the spirit of learning-augmented algorithms, we then show that if the heuristic forest found in step (1) overlaps with an optimal MST, we can approximate the original MST problem in subquadratic time, where the approximation factor depends on a measure of overlap. In practice, we find nearly optimal spanning trees for a wide range of metrics, while being orders of magnitude faster than exact algorithms.

## 1. Introduction

Finding a minimum spanning tree of a graph is a classical combinatorial problem with well-known algorithms dating back to the early and mid 1900s (Borůvka, 1926; Prim, 1957; Kruskal, 1956). A widely-studied special case in theory and practice is the *metric* MST problem, where each node corresponds to a point in a metric space and every pair of points defines an edge with weight equal to the distance between points. Finding a metric MST has widespread applications including network design (Loberman & Weinberger, 1957), approximation algorithms for traveling salesman

problems (Held & Karp, 1970), and feature selection (Labbé et al., 2023). The problem also has a very well-known connection to hierarchical clustering (Gower & Ross, 1969) and has been used as a key step in clustering astronomical data (Barrow et al., 1985; March et al., 2010), analyzing gene expression data (Xu et al., 2002), document clustering (Xu & Uberbacher, 1997), and various image segmentation and classification tasks (Xu & Uberbacher, 1997; An et al., 2000; La Grassa et al., 2022).

This paper is motivated by challenges in efficiently computing metric MSTs in modern machine learning (ML) and data mining applications. The most fundamental challenge is simply the massive size of modern datasets. In theory one can always find an MST for $n$ points by computing all $O(n^2)$ distances and running an existing MST algorithm. However, this quadratic complexity is far too expensive both in terms of runtime and memory for massive datasets. A second challenge is handling complicated metric spaces. Most existing algorithms for metric MSTs are designed for Euclidean distances or other simple metric spaces, often with a particular focus on small-length feature vectors (March et al., 2010; Agarwal et al., 1990; Shamos & Hoey, 1975; Vaidya, 1988; Arya & Mount, 2016; Wang et al., 2021). While useful in certain settings, these are limited in their applicability to high-dimensional feature spaces and complex distance functions. Many modern ML tasks focus on non-Cartesian data (e.g., videos, images, text, nodes in a graph, or even entire graphs) which must be classified, clustered, or otherwise compared, possibly after being embedded into some metric space. In these settings, even querying the distance score between two data points becomes non-trivial and often involves some level of uncertainty. Computing distances for popular non-Euclidean metrics like Levenshtein distance or graph kernels is far more expensive than computing Euclidean distances. This motivates a body of research on minimizing the number of queries needed to find a minimum spanning tree (Bateni et al., 2024; Erlebach et al., 2022; Hoffmann et al., 2008; Megow et al., 2017).

These applications and challenges motivate new approaches for efficiently finding good spanning trees for a set of points in an arbitrary metric space. Ideally, we would like an

[1]Department of Computer Science and Engineering, Texas A&M University [2]Center for Applied Scientific Computing, Lawrence Livermore National Laboratory. Correspondence to: Nate Veldt <nveldt@tamu.edu>.

*Proceedings of the 42nd International Conference on Machine Learning*, Vancouver, Canada. PMLR 267, 2025. Copyright 2025 by the author(s).

algorithm whose memory, runtime, and distance query complexity are all $o(n^2)$, while still being able to find spanning trees with strong theoretical guarantees in arbitrary metric spaces. Unfortunately, there are known hardness results that pose challenges for obtaining meaningful subquadratic algorithms. In particular, it is known that finding any constant factor approximation for an MST in an arbitrary metric space requires knowing $\Omega(n^2)$ edges in the underlying metric graph (Indyk, 1999). Overcoming this inherent challenge requires exploring additional assumptions and alternative types of theoretical approximation guarantees.

**The present work: metric forest completion.** To achieve our design goals and overcome existing hardness results, we take our inspiration from the nascent field of learning-augmented algorithms, also known as algorithms with predictions (Mitzenmacher & Vassilvitskii, 2022). The learning-augmented paradigm assumes access to an ML heuristic that provides a prediction or "warm start" that is useful in practice but does not come with a priori theoretical guarantees. The goal is to design an algorithm that (1) comes with improved theoretical guarantees when the ML heuristic performs well, and (2) recovers similar worst-case guarantees if the ML heuristic performs poorly. Improved theoretical guarantees can take various forms, including faster runtimes (e.g., for sorting (Bai & Coester, 2023), binary search (Lin et al., 2022) or maximum $s$-$t$ flows (Davies et al., 2023)), improved approximation factors (e.g., for NP-hard clustering problems (Ergun et al., 2022; Nguyen et al., 2023)), better competitive ratios for online algorithms (e.g., for ski rental problems (Shin et al., 2023)), or some combination of the above (e.g., better trade-offs for space requirements vs. false positive rates for Bloom filters (Kraska et al., 2018)). These improved guarantees are given in terms of some parameter measuring the quality of the prediction, which is typically not known in practice but provides a concrete measure of error that can be used in theoretical analysis.

In this paper we specifically assume access to fragments of a spanning tree for a metric MST problem (called the *initial forest*), that takes the form of a spanning tree for each component of some partitioning of the data objects. This input can be interpreted as a heuristic approximation for the forest that would be obtained by running a few iterations of a classical MST algorithm such as Kruskal's (Kruskal, 1956) or Boruvka's (Borůvka, 1926). Given this input, we formalize the METRIC FOREST COMPLETION problem (MFC), whose goal is to find a minimum-weight set of edges that connects disjoint components to produce a full spanning tree. Although optimally solving MFC takes $\Omega(n^2)$ distance queries, we design a subquadratic approximation algorithm for MFC and prove a learning-augmented style approximation guarantee for the original metric MST problem. To

summarize, we have the following contributions.

- **New algorithmic framework.** We introduce METRIC FOREST COMPLETION for large-scale metric MST problems, along with strategies for computing an initial forest and a discussion of how our problem fits into the recent framework of learning-augmented algorithms.

- **Approximate completion algorithm.** We prove that optimally solving MFC requires $\Omega(n^2)$ edge queries, but provide an approximation algorithm with approximation factor $\approx 2.62$. Our algorithm has a query complexity of $o(n^2)$ as long as the initial forest has $o(n)$ components.

- **Learning-augmented approximation guarantees.** We prove that if the initial forest $\gamma$-overlaps with an optimal MST, then our algorithm is a $(2\gamma + 1)$-approximation algorithm for the metric MST problem. If $\gamma = 1$, this means all edges in the forest are contained in an optimal MST. A precise definition for $\gamma > 1$ is given in Section 3.

- **Experiments.** We show that our method is extremely scalable and obtains very good results on synthetic and real datasets. We also show that simple heuristics provide initial forests with $\gamma$-overlap values that are typically smaller than 2 on various datasets and distance functions, including many non-Euclidean metrics.

## 2. Technical Preliminaries and Related Work

We cover several technical preliminaries to set the stage for our new MFC problem and algorithms. Additional related work can be found in Section 5.

**Graph notation and minimum spanning trees.** For an undirected graph $G = (V, E)$ and weight function $w \colon E \to \mathbb{R}^+$, we denote the weight of an edge $e = (u, v)$ as $w(u, v)$, $w_{uv}$, or $w_e$. The weight of an edge set $F \subseteq E$ is denoted by $w(F) = \sum_{e \in F} w_e$. We frequently use $w(G) = w(E)$ to denote the weight of all edges in $G = (V, E)$.

The minimum spanning tree problem on $G$ with respect to weight function $w$ seeks a spanning tree $T = (V, E_T)$ of $G$ that minimizes $w(E_T)$. When $w$ is clear from context we will refer to $T$ simply as an MST of $G$. We do often consider multiple spanning trees of the same graph, each optimal for a different weight function. In these cases we explicitly state the weight function associated with an MST.

There are many well-known greedy algorithms for optimally constructing an MST. We review Kruskal's (Kruskal, 1956) and Boruvka's (Borůvka, 1926) algorithms as their mechanics are relevant for understanding our approach and results. These methods first place all nodes into singleton components. Each iteration of Kruskal's algorithm identifies a minimum weight edge that connects two nodes in different components, and adds it to a forest of edges (initialized to the empty set at the outset of the algorithm) that is guaranteed to be part of some MST. This proceeds until

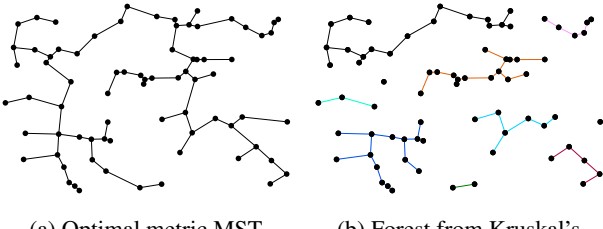

(a) Optimal metric MST     (b) Forest from Kruskal's

Figure 1: (a) An MST for an implicit graph $G_{\mathcal{X}}$ defined by 75 points in 2D Euclidean space. (b) The forest at an intermediate step of Kruskal's algorithm.

all components of the forest have been merged into one tree. The method is equivalent to ordering all edges based on weight and visiting them in order, greedily adding the $i$th edge to the spanning tree if and only if its endpoints are in different components. Boruvka's algorithm (Borůvka, 1926) is similar, but instead identifies a minimum weight edge incident to *each* component, adding all of them to the growing forest. Both algorithms can be implemented to run in $O(m \log n)$ time where $n = |V|$ and $m = |E|$. Faster algorithms, often based on one of these algorithms, have been developed. The fastest deterministic algorithm has a runtime of $O(m \cdot \alpha(m, n))$ where $\alpha$ is the inverse of the Ackerman function (Chazelle, 2000). For this paper it suffices to know that finding an MST takes $\tilde{O}(m)$ time where $\tilde{O}$ hides factors that are logarithmic (or smaller) in $n$.

**The metric MST problem.** Throughout the paper we let $(\mathcal{X}, d)$ be a finite metric space where $\mathcal{X} = \{x_1, x_2, \ldots, x_n\}$ is a set of data points and $d(x_i, x_j)$ is the distance between the $i$th and $j$th points. In order to be a metric space, $d$ must satisfy the triangle inequality: $d(x_i, x_j) \leq d(x_i, x_k) + d(x_k, x_j)$ for all triplets $i, j, k \in [n] = \{1, 2, \ldots n\}$. Aside from this we make no formal assumptions about the points or $d$. The space is associated with a complete graph $G_{\mathcal{X}} = (\mathcal{X}, E_{\mathcal{X}})$ where we treat $\mathcal{X}$ as a node set and the edge set $E_{\mathcal{X}} = \binom{\mathcal{X}}{2}$ includes all pairs of nodes. The weight function $w_{\mathcal{X}}$ for $G_{\mathcal{X}}$ is defined by $w_{\mathcal{X}}(i, j) = d(x_i, x_j)$. This graph $G_{\mathcal{X}}$ is implicit; in order to know the weight of an edge we must query the distance function $d$. We make no assumptions regarding the complexity of querying distances. Rather, we will give the complexity of our algorithms both in terms of the runtime and number of queries.

The *metric* minimum spanning tree problem is simply the MST problem applied to $G_{\mathcal{X}}$ (see Figure 1a). This can be solved by querying the distance function $O(n^2)$ times to form $G_{\mathcal{X}}$, and then applying an existing MST algorithm. We show it is more practical to implicitly deal with $G_{\mathcal{X}}$ via edge queries, while solving different types of nearest neighbor problems over subsets of $\mathcal{X}$.

**Metric MSTs and bichromatic closest pairs.** The *bichro-matic closest pair problem* (BCP) is a particularly relevant computational primitive for metric MSTs. The input to BCP is two sets of points $A$ and $B$ in a metric space. The goal is then to find a pair of opposite-set points (one from $A$ and one from $B$) with smallest distance. One can implicitly apply a classical MST algorithm such as Kruskal's or Boruvka's algorithm to $G_{\mathcal{X}}$ by repeatedly solving BCP problems. In more detail, these algorithms must find one or more disconnected components in an intermediate forest (see Figure 1b) to join at each step via minimum weight edges. Finding a minimum weight edge between two components exactly corresponds to a BCP problem. This connection between metric MSTs and BCP is well known and has been leveraged in many prior works on metric MSTs (Agarwal et al., 1990; Callahan & Kosaraju, 1993; Narasimhan & Zachariasen, 2001; Chatterjee et al., 2010).

## 3. Metric Forest Completion

We now formalize our METRIC FOREST COMPLETION (MFC) framework, which assumes access to an initial forest that is then grown into a full spanning tree.

**Defining the initial forest.** As a starting point for the metric MST problem on $(\mathcal{X}, d)$, we assume access to a partitioning $\mathcal{P} = \{P_1, P_2, \ldots, P_t\}$ where $\mathcal{X} = \bigcup_{i=1}^{t} P_i$ and $P_i \cap P_j = \emptyset$ for $i \neq j$. For each component $P_i$ we have a partition spanning tree $T_i = (P_i, E_{T_i})$ for that component. See Figure 2a for an illustration. Let $G_t = (\mathcal{X}, E_t)$ represent the union of these trees, which has the same node set as $G_{\mathcal{X}}$, and edge set $E_t = \bigcup_{i=1}^{t} E_{T_i}$. Each set $P_i$ for $i \in [t]$ defines a group of points in $\mathcal{X}$ as well as a connected component of $G_t$. We refer to this as the *initial forest* for MFC. To provide intuition, the initial forest can be viewed as a proxy for the forest obtained at an intermediate step of Kruskal's or Boruvka's algorithm (see Figure 1b). While this serves as a useful analogy, we stress that the partitioning will typically be obtained using much cheaper methods and will not satisfy any formal approximation guarantees.

Our theoretical analysis requires very few assumptions about the partitioning $\mathcal{P}$ and partition spanning trees (initial forest) $\{T_i \colon i = 1, 2, \ldots t\}$ that are given as input to the METRIC FOREST COMPLETION problem. In particular, we need not assume that $T_i$ is a minimum spanning tree or even a good spanning tree of $P_i$. Similarly, the components are not required to be sets of points that are close together in the metric space in order for the problem to be well-defined. Nevertheless, *in practice* the hope is that $\mathcal{P}$ identifies groups of nearby points in $\mathcal{X}$ and that $T_i$ is a reasonably good spanning tree for $P_i$ for each $i \in [t] = \{1, 2, \ldots, t\}$. Appendix A covers two practical strategies and corresponding runtimes for finding an initial forest. In summary these are:

*Strategy 1: $k$-centering.* Apply a fast $k$-centering heuristic

to $\mathcal{X}$ to form $\mathcal{P}$, e.g., using a simple 2-approximation (Gonzalez, 1985) or fast distributed methods (Malkomes et al., 2015; McClintock & Wirth, 2016).

*Strategy 2: $k$-NN graph.* Compute an approximate $k$-nearest neighbor graph of $\mathcal{X}$ for a reasonably small $k$, e.g., via the scalable $k$-NN descent algorithm (Dong et al., 2011) or its distributed generalization (Iwabuchi et al., 2023).

Similar but more restrictive strategies have been used by previous heuristics for Euclidean MSTs (see Section 5).

**Defining the MFC problem.** METRIC FOREST COMPLETION seeks to connect an initial forest into a spanning tree for $G_{\mathcal{X}}$. Let $P(x) \in \mathcal{P}$ denote the component that $x \in \mathcal{X}$ belongs to. The set of inter-component edges is

$$\mathcal{I} = \{(x, y) \in \mathcal{X} \times \mathcal{X} : P(x) \neq P(y)\}.$$

We wish to find a minimum weight set of edges $M \subseteq \mathcal{I}$ so that $M \cup E_t$ defines a connected graph on $\mathcal{X}$. If $M$ satisfies these constraints we say it is a valid *completion set* and that $M$ *completes* $\mathcal{P}$. The MFC problem can then be written as

$$\begin{aligned} \text{minimize} \quad & w_{\mathcal{X}}(M) + w_{\mathcal{X}}(E_t) \\ \text{subject to} \quad & M \text{ completes } \mathcal{P}. \end{aligned} \quad (1)$$

Let $M^*$ denote an optimal completion set. The graph $T^* = (\mathcal{X}, E_t \cup M^*)$ is then guaranteed to be a tree (see Figure 2b); if not we could remove edges to decrease the weight while still spanning $\mathcal{X}$. If the forest $G_t$ is in fact contained in some optimal spanning tree of $G_{\mathcal{X}}$ (which would be the case if it were obtained by running a few iterations of Kruskal's or Boruvka's algorithm), then solving MFC would produce an MST of $G_{\mathcal{X}}$. In practice this will typically not be the case, but the problem remains well-defined regardless of any assumptions about the quality of the initial forest.

The objective function in Eq. (1) includes the weight of the initial forest $w_{\mathcal{X}}(E_t)$. Although this is constant with respect to $M$ and does not affect optimal solutions, there are several reasons to incorporate this term explicitly. Most importantly, our ultimate goal is to obtain a good spanning tree for all of $G_{\mathcal{X}}$, and thus the weight of the full spanning tree (i.e., the objective in Eq. 1) is a more natural measure. Considering the weight of the full spanning tree also makes more sense in the context of our learning-augmented algorithm analysis, where the goal is to approximate the original metric MST problem on $G_{\mathcal{X}}$, under different assumptions about the initial forest. We note finally that excluding the term $w_{\mathcal{X}}(E_t)$ rules out the possibility of any meaningful approximation results. We prove the following result using a reduction from BCP to MFC, combined with a slight variation of a simple lower bound for monochromatic closest pair that was shown in Section 9 of Indyk (1999).

**Theorem 3.1.** *Every optimal algorithm for MFC has $\Omega(n^2)$ query complexity. Furthermore, for any multiplicative factor*

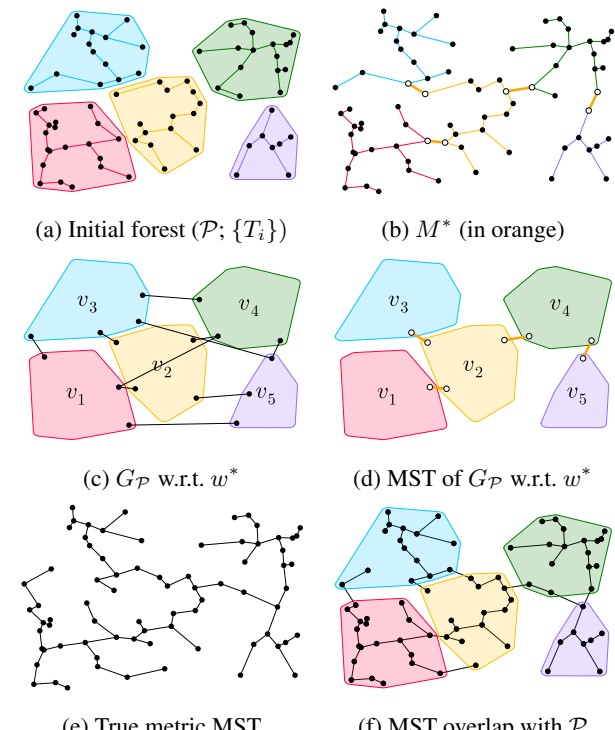

(a) Initial forest $(\mathcal{P}; \{T_i\})$     (b) $M^*$ (in orange)

(c) $G_{\mathcal{P}}$ w.r.t. $w^*$     (d) MST of $G_{\mathcal{P}}$ w.r.t. $w^*$

(e) True metric MST     (f) MST overlap with $\mathcal{P}$

Figure 2: (a) An initial forest for a small metric MST problem. (b) The optimal solution to MFC. (c) The coarsened graph $G_{\mathcal{P}}$ with respect to optimal weight function $w^*$. (d) An MST of $G_{\mathcal{P}}$ with respect to $w^*$, which can in theory be used to find $M^*$. (e) A true MST of the data points. (f) The overlap between the initial forest and this true MST, which can be used to certify $\gamma \leq 1.12$ for this example.

$p \geq 1$ *(not necessarily a constant), any algorithm that finds a set $M \subseteq \mathcal{I}$ that is feasible for (1) and satisfies $w_{\mathcal{X}}(M) \leq p \cdot w_{\mathcal{X}}(M^*)$ requires $\Omega(n^2)$ queries.*

*Proof of Theorem 3.1.* Let $\mathcal{X}$ be a set of $n$ points that are partitioned into two sets $P_1$ and $P_2$ of size $n/2$. Define a distance function $d$ such that $d(a, b) = 1$ for a randomly chosen pair $(a, b) \in P_1 \times P_2$, and such that $d(x, y) = 2p$ for all other pairs $(x, y) \in \binom{\mathcal{X}}{2} \backslash \{(a, b)\}$. Note that this $d$ is a metric. The MFC problem on this instance is identical to solving BCP on $P_1$ and $P_2$. The unique optimal solution is exactly $M^* = (a, b)$, and no other choice of $M \subseteq \mathcal{I}$ comes within a factor $p$ of this solution. Thus, any $p$-approximation algorithm must find the pair $(a, b)$ with distance 1 among a collection of $\Omega(n^2)$ pairs, where all other pairs have distance $2p$. This requires $\Omega(n^2)$ queries. $\quad\square$

**The MFC coarsened graph.** The MFC problem is equivalent to finding a minimum spanning tree in a *coarsened graph* $G_{\mathcal{P}} = (V_{\mathcal{P}}, E_{\mathcal{P}})$ with node set $V_{\mathcal{P}} = \{v_1, v_2, \ldots, v_t\}$ where $v_i$ represents component $P_i$. We

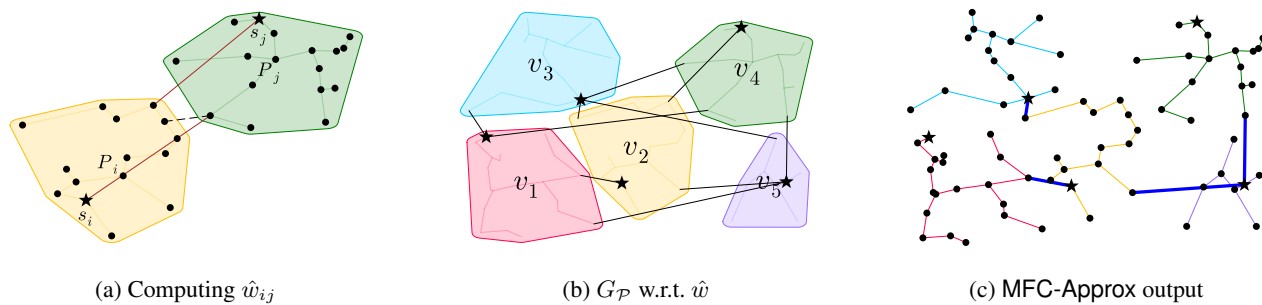

(a) Computing $\hat{w}_{ij}$       (b) $G_{\mathcal{P}}$ w.r.t. $\hat{w}$       (c) MFC-Approx output

Figure 3: (a) Finding the minimum distance between components $P_i$ and $P_j$ (dashed line) is an expensive bichromatic closest pair problem. MFC-Approx instead performs a cheaper nearest neighbor query for a *representative* point in each component ($s_i$ and $s_j$, shown as stars). (b) Applying this to each pair of components produces a weight function $\hat{w}$ for the coarsened graph $G_{\mathcal{P}}$. Finding an MST of $G_{\mathcal{P}}$ with respect to $\hat{w}$ yields (c) a 2.62-approximation for MFC.

refer to $v_i$ as the $i$th *component node*. This graph is complete: $E_{\mathcal{P}}$ includes all pairs of component nodes. Figure 2c provides an illustration of the coarsened graph. Finding an MFC solution $M^* \subseteq \mathcal{I}$ is equivalent to finding an MST in $G_{\mathcal{P}}$ (see Figure 2d) with respect to the weight function $w^* : E_{\mathcal{P}} \to \mathbb{R}^+$ defined for every $i, j \in \{1, 2, \ldots, t\}$ by

$$w_{ij}^* = w^*(v_i, v_j) = d(P_i, P_j) = \min_{x \in P_i, y \in P_j} d(x, y). \quad (2)$$

Computing $w_{ij}^*$ exactly requires solving a bichromatic closest pair problem over sets $P_i$ and $P_j$. A straightforward approach for computing $w_{ij}^*$ is to check all $|P_i| \cdot |P_j|$ pairs of points in $P_i \times P_j$. However, in the worst case where cluster sizes are balanced, computing all of $w^*$ requires $\Omega(n^2)$ queries, which is not surprising given Theorem 3.1. Nevertheless, this notion of a coarsened graph will be very useful in developing approximation algorithms for MFC.

**MFC as a learning-augmented framework.** Our approach fits the framework of learning-augmented algorithms in that the initial forest can be viewed as a prediction for a partial metric MST, such as the forest obtained by running several iterations of a classical MST algorithm. In an ideal setting, the initial forest would be a subset of an optimal MST. If so, then an optimal solution to MFC would produce an optimal metric MST. We relax this by introducing a more general way to measure how much an optimal MST "overlaps" with initial forest components. Let $\mathcal{T}_{\mathcal{X}}$ represent the set of minimum spanning trees of $G_{\mathcal{X}}$, and $T \in \mathcal{T}_{\mathcal{X}}$ denote an arbitrary MST. For components $\mathcal{P} = \{P_1, P_2, \ldots, P_t\}$, let $T(P_i)$ denote the induced subgraph of $T$ on $P_i$, and let $T(\mathcal{P}) = \bigcup_{i=1}^{t} T(P_i)$ denote the edges of $T$ contained inside components of $\mathcal{P}$. Larger values of $w_{\mathcal{X}}(T(\mathcal{P}))$ indicate better initial forests, since this means an optimal MST places a larger weight of edges inside these components. We define the $\gamma$-overlap for the initial forest to be the ratio

$$\gamma(\mathcal{P}) = w_{\mathcal{X}}(E_t) \Big/ \max_{T \in \mathcal{T}_{\mathcal{X}}} w_{\mathcal{X}}(T(\mathcal{P})). \quad (3)$$

This measures the weight of edges that the initial forest places inside $\mathcal{P}$, relative to the weight of edges an optimal MST places inside $\mathcal{P}$. When $\mathcal{P}$ is clear from context we will simply write $\gamma$. Lower ratios for $\gamma$ are better. Because of the minimizing property of any MST, and because $E_t$ is fully connected within each $P_i$ even though an MST need not be, we always have $\gamma(\mathcal{P}) \geq 1$. This bound is tight exactly when the initial forest is contained in some optimal MST. For a reasonable initial forest where each $P_i$ is a set of nearby points and $T_i$ is a reasonably good spanning tree for $P_i$, we would expect $\gamma$ to be larger than 1 but still not too large. Figure 2f provides an example where we can certify that $\gamma \leq 1.12$ by comparing against one optimal MST. We later show experimentally that we can quickly obtain initial forests with small $\gamma$-overlap for a wide range of datasets and metrics. In practical applications we typically would not compute $\gamma$, as this is more computationally expensive that solving the original metric MST problem. However, using $\gamma$ as a theoretical measure of quality, we will design a learning-augmented algorithm that improves on worst-case results when the quality is good, while still recovering standard worst-case results when the quality is bad.

## 4. Approximate Completion Algorithm

We now present an algorithm that approximates MFC to within a factor $c < 2.62$. We also prove it can be viewed as a learning-augmented algorithm for metric MST, where the approximation factor depends on the $\gamma$-overlap of the initial forest. Pseudocode for our algorithm is provided in the appendix. Here in the main text we give a full description of the algorithm along with visual aids in Figure 3. Due to space constraints, proofs are relegated to the appendix.

**Algorithm description.** Our algorithm starts by choosing an arbitrary point $s_i \in P_i$ for each $i \in \{1, 2, \ldots, t\}$ to be the component's *representative* (starred nodes in Figure 3). The algorithm computes the distance between every point $x \in \mathcal{X}$ and all of the other representatives. For every pair of

distinct components $i$ and $j$ we compute the weights:

$$w_{i \to j} = \min_{x_i \in P_i} d(x_i, s_j) \quad \text{(the closest } P_i \text{ node to } s_j\text{)} \quad (4)$$

$$w_{j \to i} = \min_{x_j \in P_j} d(x_j, s_i) \quad \text{(the closest } P_j \text{ node to } s_i\text{)}. \quad (5)$$

We then define the approximate edge weight between component nodes $v_i$ and $v_j$ to be:

$$\hat{w}_{ij} = \min\{w_{i \to j}, w_{j \to i}\}. \quad (6)$$

This upper bounds the minimum distance $w_{ij}^*$ between the two components (Figure 3a). Computing this for all pairs of components creates a new weight function $\hat{w}$ for the coarsened graph $G_\mathcal{P}$ (Figure 3b). The algorithm keeps track of the points in $\mathcal{X}$ that define these edge weights in $G_\mathcal{P}$. It then computes an MST in $G_\mathcal{P}$ with respect to $\hat{w}$, then identifies the corresponding edges in $G_\mathcal{X}$, to produce a feasible solution for MFC (Figure 3c). We refer to this algorithm as MFC-Approx.

**Approximation guarantees.** For our analysis we separate edges of the coarsened graph $G_\mathcal{P} = (V_\mathcal{P}, E_\mathcal{P})$ into two categories. For an arbitrary constant $\beta > 1$, edge $(v_i, v_j) \in E_\mathcal{P}$ is $\beta$-*bounded* if $\hat{w}_{ij} \leq \beta w_{ij}^*$, otherwise it is $\beta$-*unbounded*. The following Lemma bounds the weight of $\beta$-unbounded edges in terms of the initial forest.

**Lemma 4.1.** *Let $\beta > 1$ be arbitrary. If $\hat{w}_{ij} > \beta w_{ij}^*$, then $\hat{w}_{ij} < \min\{w_\mathcal{X}(T_i), w_\mathcal{X}(T_j)\} \cdot \beta/(\beta - 1)$.*

Lemma 4.1 aids in proving the following guarantee.

**Theorem 4.2.** *MFC-approx is a $\beta$-approximation algorithm for MFC for $\beta = (3 + \sqrt{5})/2 < 2.62$.*

The key proof idea is to separately bound the weight of $\beta$-bounded and $\beta$-unbounded edges. If MFC-Approx includes a $\beta$-bounded edge in its MST of $G_\mathcal{P}$, the weight of this edge is bounded in terms of the optimal edge weights $w^*$. Our analysis then uses Lemma 4.1 to bound the cost of $\beta$-unbounded edges in terms of the weight of the initial forest. The approximation factor in Theorem 4.2 is obtained by choosing a value of $\beta$ that best balances these two types of costs. While this provides rough intuition, the full proof requires many additional details and some other supporting results, in order to ensure we can properly bound the weight of all $\beta$-bounded and $\beta$-unbounded edges in the resulting spanning tree at once. Appendix B provides full details. With a similar proof technique, we prove MFC-Approx is a learning-augmented algorithm for metric MST whose performance depends on the quality of the initial forest.

**Theorem 4.3.** *Let $G_\mathcal{X}$ be an implicit metric graph and $\mathcal{P}$ be an initial partitioning with $\gamma$-overlap $\gamma = \gamma(\mathcal{P})$. MFC-Approx returns a spanning tree of $G_\mathcal{X}$ that approximates the optimal metric MST to within a factor $\beta = \frac{1}{2}\left(2\gamma + 1 + \sqrt{4\gamma + 1}\right) \leq 2\gamma + 1$.*

The total runtime of MFC-Approx is $\tilde{O}(nt\mathcal{Q}_\mathcal{X} + t^2)$ where $\mathcal{Q}_\mathcal{X}$ is the complexity for one distance query in $\mathcal{X}$. This is subquadratic as long as $t\mathcal{Q}_\mathcal{X} = o(n)$. When $\mathcal{Q}_\mathcal{X} = \tilde{O}(1)$, the memory, runtime, and query complexity are all subquadratic as long as $t = o(n)$. We emphasize that this is the runtime just for the MFC problem, and does not incorporate the time it takes to compute an initial forest. Appendix C provides a more detailed runtime analysis that also considers one strategy for finding initial forests. The appendix also addresses ways to improve the algorithm in practice with little to no change in approximation guarantee.

## 5. Related Work

The most relevant related work is discussed within context throughout the manuscript. Here we discuss connections to other relevant work on MST algorithms.

**MST algorithms with predictions.** Our work shares similarities, though ultimately orthogonal goals and results, with other work on learning-augmented and query-minimizing algorithms for MSTs. Erlebach et al. (2022) assume access to (possibly erroneous) predictions for each edge weight in a graph (not necessarily a metric graph) and the goal is to minimize the number of (non-erroneous) edge-weight queries in order to compute an exact MST. Berg et al. (2023) considered an online setting where edge-weight predictions are all provided a priori but true weights are only revealed in an online fashion. After revealing a true weight, an irrevocable decision must be made about including or excluding the edge from a spanning tree. Bateni et al. (2024) considered MST computations in metric graphs, in settings where one has access both to a weak oracle (providing a similar type of edge-weight prediction) and a strong oracle giving true distances. They focus on bounding the number of strong oracle queries needed to find an exact or approximate MST. These prior works share similarities with our work in their goal to minimize certain types of queries. However, they differ in that the learning-augmented information takes the form of edge-weight predictions, rather than an initial forest. These works also perform $\Omega(n^2)$ queries in the worst case, which is prohibitive for large $n$. Our work is distinctive in its focus on subquadratic algorithms for approximate MSTs.

**Algorithms for metric MST.** Many previous papers focus on improving algorithmic guarantees for variants of the metric MST problem, especially for the special case of Euclidean distances. For $d$-dimensional Euclidean space when $d = O(1)$, an exact MST can be computed in $O(n^{2-2/(\lceil d/2 \rceil + 1) + \varepsilon})$ time (Agarwal et al., 1990), and a $(1 + \varepsilon)$-approximate solution can be found in time $O(n \log n + (\varepsilon^{-2} \log^2 \frac{1}{\varepsilon})n)$ time, where the big-$O$ notation hides constants of the form $O(1)^d$ (Arya & Mount, 2016). For high-dimensional spaces, there are also known

approaches for obtaining $c$-approximate Euclidean MSTs where subquadratic runtimes depend on the value of the desired approximation factor $c > 1$ (Har-Peled et al., 2013; 2012). There are also many recent improved theoretical results for parallel and streaming variants of the metric MST problem (Jayaram et al., 2024; Chen et al., 2022; 2023; Wang et al., 2021; Azarmehr et al., 2025; March et al., 2010), most of which again focus on the Euclidean case.

**Partitioning heuristics for MSTs.** There are several existing divide-and-conquer techniques for finding an approximate Euclidean MST (Chen, 2013; Zhong et al., 2015; Mishra & Mohanty, 2020; Jothi et al., 2018). Similar to our approach, these methods begin by partitioning the data into smaller components, using methods such as $k$-means (Zhong et al., 2015; Jothi et al., 2018), finding components of a $k$-NN graph (Chen, 2013), or using recursive partitioning techniques (Mishra & Mohanty, 2020). An MST for the entire dataset is obtained using various heuristics for connecting components internally and then connecting disjoint components. This often involves computing an MST on some form of coarsened graph as a substep. Despite high-level similarities, these methods differ from our forest completion framework in that they focus exclusively on Euclidean space, an assumption that is essential for the partitioning schemes used and the techniques for connecting components. The other major difference is that these approaches do not attempt to provide any type of approximation guarantee, which is our main focus.

**Alternate uses of the acronym MFC.** Liu et al. (2014) used MFC to denote a feature selection technique they called *MST-based Feature Clustering*. This involves computing an MST, but does not involve any notion of forest completion. Kor et al. (2011) studied distributed algorithms for verifying whether a tree is a minimum spanning tree of a graph. Their problem assumes an input graph is partitioned among many processors, and the candidate tree consists of marked edges that are partitioned into a so-called *MST fragment collection* (MFC). These fragments bear a cursory resemblance with the forest used as input to METRIC FOREST COMPLETION problem, but Kor et al. (2011) do not focus on metric graphs or completing initial forests. Furthermore, their problem and approach rely on evaluating a full graph in memory, which is intractable in our setting.

## 6. Experiments

We run a large number of numerical experiments on synthetic and real-world datasets to show the practical utility of our MFC framework. The overall utility of our approach relies both on the performance of MFC-Approx as well as our ability to obtain good initial forests. We therefore evaluate both aspects in practice. Our experiments are designed to address the following questions:

**Q1**. How does this framework perform in terms of runtime and spanning tree cost?

**Q2**. What $\gamma$-overlap (see Eq. 3) can be achieved in practice by scalable partitioning heuristics?

**Q3**. How does practical performance compare with the theoretical bound in Theorem 4.3?

**Q4**. How do differences in the structure of the data and choice of distance metric affect performance?

**Implementation details and experimental setup.** We implemented our algorithms in templated C++ code which allows for easy specialization of different distance metrics. Experiments were run on a research server with two AMD EPYC 7543 32-Core Processors and 1 TB of RAM running Ubuntu 20.04.1. The code was compiled with clang version 20.0.0 with O3 using libc++ version 20.0.0.

To generate an initial forest, we partition data points using a simple $k$-center algorithm and compute an exact MST for each component using Kruskal's algorithm. The runtimes we report include the time it takes to compute an initial forest with this approach. See Appendix C.2 for details on how this initial forest computation affects runtimes. We compare against an exact MST for all of $\mathcal{X}$, computed by generating all $\binom{n}{2}$ edge weights then running Kruskal's algorithm. We evaluate our framework in terms of runtime and cost ratio (i.e., the spanning tree cost divided by the weight of an optimal MST). We also compute a bound $\bar{\gamma} \geq \gamma(\mathcal{P})$ on $\gamma$-overlap for our initial forest, obtained by computing the overlap between the initial forest and the one optimal MST we compute for our comparisons. Appendix D provides additional details about our experimental setup, design choices, and evaluation techniques.

**Uniform random data.** We run a large number of experiments on uniform random data in $d$-dimensional Euclidean space to illustrate the strong performance of our MFC framework in terms of runtime and spanning tree quality (addressing **Q1** and **Q2**) in a simple controlled setting. Figure 4 shows results for $d \in \{8, 256\}$ as $n$ increases across a range of choices for component number $t$. See Appendix D for results on a wider range of dimensions $d$.

As $t$ increases, we see improvements in asymptotic speedups over the exact baseline algorithm (up to a 300x speedup for $d = 8$ and 30x speedup for $d = 256$). As $t$ increases the quality of the approximate spanning tree also decreases slightly, but the cost approximation ratio still remains very good (i.e., close to 1) in all cases. The $\gamma$-overlap bound $\bar{\gamma}$ tend to be small for low dimensions, but gets large as $d$ increases (e.g., $\bar{\gamma}$ close to 30 for $d = 256$). For a given $\bar{\gamma}$, Theorem 4.3 guarantees roughly a $(2\bar{\gamma} + 1)$-approximation. For $d = 8$ this approximation ranges from 3 to 6, while for $d = 256$ it ranges from 3 to around 60. Despite this, the cost ratios obtained remain very close to 1, and even improve

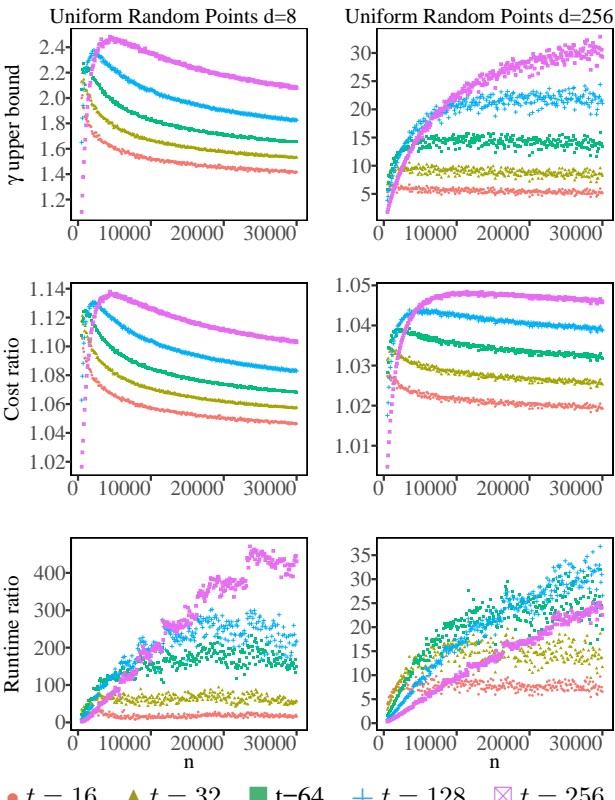

Figure 4: Results on synthetic uniform random data for dimensions $d \in \{8, 256\}$. Each point represents the average of 16 samples of the data for a fixed $n$ and choice of component number $t$ for our framework. Runtime ratio is the ratio between the runtime of the optimal MST divided by the runtime of our method. See main text for discussion.

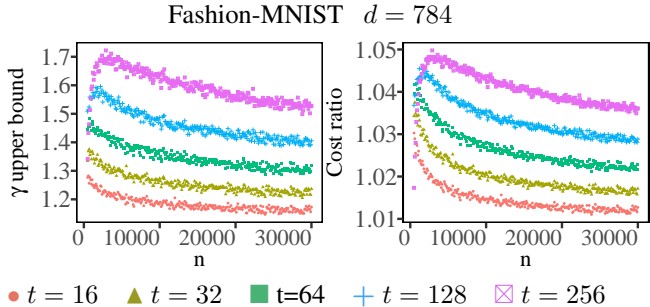

Figure 5: Results for Fashion-MNIST. Each point is the average of 16 samples for fixed $n$ and $t$.

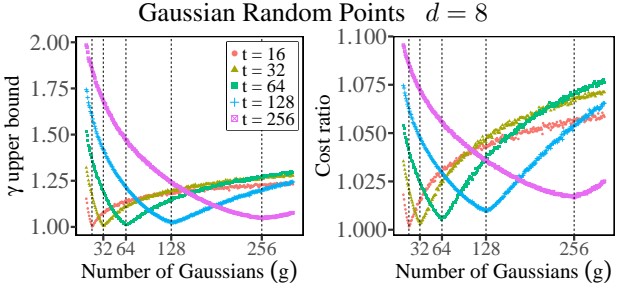

Figure 6: Results on synthetic data with clustering structure: mixtures of $g$ Gaussians. Cost ratio and $\gamma$-overlap bound $\bar{\gamma}$ is minimized when $g$ roughly matches the number of components $t$ in our MFC framework.

slightly as $d$ increases. This tells us first of all that in actual numerical experiments, our approach greatly exceeds our theoretical bounds (addressing **Q3**). We conjecture that for these high-dimensional point clouds (which lack any underlying structure), there are a large number of spanning trees that are *nearly* optimal in terms of weight but are structurally very different from an optimal spanning tree (which may be unique). This could lead to high $\gamma$ values despite our good cost ratios.

**Improved results on clustered data.** Although $\bar{\gamma}$ is large for high-dimensional uniform random data, the lack of structure in this synthetic data is atypical for most applications. For example, metric spanning trees are often computed as the first step in clustering data. Therefore, as part of our answers to Questions **Q2** (on $\gamma$ values) and **Q4** (on the structure of the data), we explore how our framework performs on datasets with some level of clustering structure. We begin by running a similar set of experiments as in Figure 4, but we instead sample points uniformly at random from the Fashion-MNIST dataset, where points represent images of

clothing items. Although images do not perfectly cluster into different classes, there is still clear underlying structure (e.g., we expect images of sneakers to look different from images of trousers). Figure 5 shows that for this dataset, our MFC framework achieves slightly better cost ratios and far better $\gamma$-overlap bounds than for uniform random data (Figure 4), even though the dimension of Fashion-MNIST points ($d = 784$) is much larger than the dimensions we considered for uniform data. We also observed similar trends in runtime improvements as in Figure 4 (see Appendix D).

We further examine the effect of clustering structure in a controlled setting by generating mixtures of $g$ Gaussians in 8-dimensional Euclidean space, where parameters are selected to produce good (even if imperfect) clustering structure. For each Gaussian we generate $\lfloor 20000/g \rfloor$ points, so that $n \approx 20000$ for each dataset. Figure 6 shows $\bar{\gamma}$ values and cost ratios for a range of component numbers $t$ as $g$ varies. Significantly, the quality of our spanning trees gets better and better as the number of components $t$ gets closer to the number of Gaussians $g$ (i.e., true number of clusters in the data). This is illustrated by sharp valleys in the plots for $\bar{\gamma}$ and cost ratio when $t \approx g$. Furthermore, even when $t$ and $g$ are not close to each other, $\bar{\gamma}$ values and cost ratios are still smaller than the ones obtained on uniform random data with the same dimension $d = 8$ (see Figure 4). This

Table 1: Results on real-world datasets.

| Dataset | | OPT | $t$ | | |
|---|---|---|---|---|---|
| | | | 16 | 32 | 128 |
| *Cooking* | $\bar{\gamma}$ | - | 1.77 | 2.01 | 2.42 |
| | Cost Ratio | 1 | 1.04 | 1.05 | 1.07 |
| | Run Ratio | 1 | 4.39 | 6.52 | 9.97 |
| $n \approx 40k$ | Run (min) | 23.1 | 5.3 | 3.5 | 2.3 |
| *Movie* | $\bar{\gamma}$ | - | 1.40 | 1.51 | 1.95 |
| | Cost Ratio | 1 | 1.01 | 1.01 | 1.02 |
| | Run Ratio | 1 | 4.03 | 5.01 | 9.34 |
| $n \approx 33k$ | Run (min) | 484.2 | 120.3 | 96.6 | 51.9 |
| *Kosarak* | $\bar{\gamma}$ | - | 1.29 | 1.74 | 2.79 |
| | Cost Ratio | 1 | 1.01 | 1.01 | 1.03 |
| | Run Ratio | 1 | 2.68 | 5.99 | 18.17 |
| $n \approx 32k$ | Run (min) | 182.3 | 68.0 | 30.4 | 10.0 |
| *Names* | $\bar{\gamma}$ | - | 1.02 | 1.06 | 1.22 |
| | Cost Ratio | 1 | 1.01 | 1.02 | 1.05 |
| | Run Ratio | 1 | 0.92 | 0.95 | 1.03 |
| $n = 30k$ | Run (min) | 2.0 | 2.1 | 2.1 | 1.9 |
| *GG-unalign.* | $\bar{\gamma}$ | - | 1.39 | 1.46 | 1.40 |
| | Cost Ratio | 1 | 1.09 | 1.10 | 1.07 |
| | Run Ratio | 1 | 2.56 | 3.27 | 2.12 |
| $n = 2.5k$ | Run (min) | 239.3 | 98.0 | 74.9 | 113.2 |
| *GG-aligned* | $\bar{\gamma}$ | - | 1.22 | 1.37 | 1.51 |
| | Cost Ratio | 1 | 1.07 | 1.12 | 1.15 |
| | Run Ratio | 1 | 1.60 | 2.38 | 5.85 |
| $n = 30k$ | Run (min) | 33.1 | 22.1 | 15.1 | 6.7 |
| *Fashion* | $\bar{\gamma}$ | - | 1.17 | 1.24 | 1.41 |
| | Cost Ratio | 1 | 1.01 | 1.02 | 1.03 |
| | Run Ratio | 1 | 4.67 | 7.18 | 12.13 |
| $n = 30k$ | Run (min) | 11.4 | 2.7 | 1.8 | 1.0 |

indicates that clustering structure helps our framework even when the number of underlying clusters is unknown. See Appendix D for additional plots, including runtimes.

**Additional results on real data.** We continue to address **Q4** by running experiments on 7 real-world datasets and corresponding metrics, which have often been used as benchmarks for similarity search algorithms and clustering algorithms. Cooking (Kaggle, 2015; Amburg et al., 2020), Kosarak (Bodon; Aumueller et al., 2024), and MovieLens (*Movie*) (Harper & Konstan, 2015; Aumueller et al., 2024) are set data so we use Jaccard distance. Names (Remy, 2021) and GreenGenes-Unaligned (*GG-unalign.*) (DeSantis et al., 2006) contain variable-length strings so we use Levenshtein edit distance. GreenGenes-Aligned (*GG-align*) (DeSantis et al., 2006) consists of fixed-length strings (all with 7682 characters) so we use Hamming distance. For Fashion-MINST (Xiao et al., 2017) we use Euclidean distance. See Appendix D for additional details on datasets.

Table 1 displays cost ratios, runtimes, and $\gamma$-overlap bound $\bar{\gamma}$ for our MFC framework. For some datasets, we are unable to compute the exact minimum spanning tree in a reasonable amount of time and space, so we subsample the data 16 times for a fixed value of $n$ and report mean results (see Appendix D for standard deviations). The $\bar{\gamma}$ values tend to be very good in practice (nearly always below 2), reinforcing the observation that our framework and theoretical guarantees are better when there is structure in the data.

In some cases, runtimes get worse as $t$ increases too much, but this matches expectations and can be guarded against in practice. For example, for the GreenGenes-Unaligned (*GG-unalign*) dataset we consider small subsets of size $n = 2500$ since data objects are long strings, and computing Levenshtein distances is expensive. When $t = 256$, the average partition size for the initial forest is roughly 10 points. In this case, the MFC step is extremely expensive. Our framework is designed for situations where $t$ is asymptotically much smaller than $n$, hence Table 1 illustrates the decrease in performance we would naturally expect when $t$ is too large. Another observation is that we only see runtime improvements for the *Names* dataset for larger $t$. It turns out this is because of extreme outliers in the *Names* dataset causing our clustering approach to create one large cluster containing most names. This can be guarded against by improving the initial clustering on the data. Appendix D provides additional details, including a breakdown of the time spent on each step of the MFC framework.

## 7. Conclusions and Discussion

We have presented a new framework for finding spanning trees in arbitrary metric spaces, which is highly scalable and grounded in rigorous approximation guarantees. This framework is based on completing an initial forest, which can be obtained efficiently using practical heuristics. This paper focuses on serial implementations and theoretical guarantees, as our framework already provides many advantages in this setting. At the same time, our work is strongly motivated by massive-scale clustering applications that require high-performance computing capabilities, and the algorithm we developed is highly parallelizable. A natural direction for further research is to develop parallel versions of our algorithm that can be run on a much larger scale. There are also many remaining questions in the serial setting. One direction is to try to improve on the $(\sqrt{5} + 3)/2$-approximation guarantee while still using subquadratic time, or obtain an approximation with better dependence on the $\gamma$-overlap parameter. Another direction is to prove lower bounds for the best possible approximation guarantees for subquadratic algorithms. There are also many opportunities to explore more efficient and practical methods for obtaining the initial forest that serves as the input to the MFC problem.

## Acknowledgements

This work was performed under the auspices of the U.S. Department of Energy by Lawrence Livermore National Laboratory under Contract DE-AC52-07NA27344 (LLNL-CONF-2002129), and was supported by LLNL LDRD project 24-ERD-024.

## Impact Statement

This paper presents work whose goal is to advance the field of Machine Learning. There are many potential societal consequences of our work, none of which we feel must be specifically highlighted here.

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

## A. Strategies for Finding an Initial Forest

In order for our approach to be meaningful for large-scale metric MST problems, we must be able to obtain a reasonably good initial forest without this dominating our overall algorithmic pipeline. Here we discuss two specific strategies for initial forest computations, both of which are fast, easy to parallelize, and motivated by techniques that are already being used in practice in large-scale clustering pipelines. In particular, there are already a number of existing heuristics for finding MSTs and hierarchical clusters that rely in some way on partitioning an initial dataset and then connecting or merging components (Zhong et al., 2015; Jothi et al., 2018; Mishra & Mohanty, 2020; Chen, 2013). These typically focus only on point cloud data, do not apply to arbitrary metric spaces, and do not come with any type of approximation guarantee. Nevertheless, they provide examples of fast heuristics for large-scale metric spanning tree problems, and serve as motivation for our more general strategies below.

**Strategy 1: Components of a $k$-NN graph.** A natural way to obtain an initial forest for $G_\mathcal{X}$ is to compute an approximate $k$-nearest neighbor graph for a reasonably small $k$, which can be accomplished with the $k$-NN descent algorithm (Dong et al., 2011) or a recent distributed generalization of this method (Iwabuchi et al., 2023). This efficiently connects a large number of points using small-weight edges. The $k$-NN graph will often be disconnected, and we can use the set of connected components as our initial components $\mathcal{P} = \{P_1, P_2, \ldots, P_t\}$. The exact number of components will depend on the distribution of the data and the number of nearest neighbors computed. For larger values of $k$, the $k$-NN graph is more expensive to compute, but then there are fewer components to connect, so there are trade-offs to consider. The components of the $k$-NN graph will typically not be trees but will be sparse (the average degree is $O(k)$), so we can use classical MST algorithms to find spanning trees for all components in $\sum_{i=1}^{t} \tilde{O}(k \cdot |P_i|) = \tilde{O}(kn)$ time. The exact runtime of the $k$-NN descent algorithm depends on various parameters settings, but prior work reports an empirical runtime of $O(n^{1.14})$ (Dong et al., 2011), with strong empirical performance across a range of different metrics and dataset sizes. We remark that $k$-NN computations have already been used elsewhere as subroutines for large-scale Euclidean MST computations (Almansoori et al., 2024; Chen, 2013).

**Strategy 2: Fast clustering heuristics.** Another approach is to form components $\mathcal{P} = \{P_1, P_2, \ldots, P_t\}$ by applying a fast clustering heuristic to $\mathcal{X}$ such as a distributed $k$-center algorithm (Malkomes et al., 2015; McClintock & Wirth, 2016). Even the simple sequential greedy 2-approximation algorithm for $k$-center can produce an approximate clustering using $O(nk)$ queries (Gonzalez, 1985). Approximate or exact minimum spanning trees for each $P_i$ can be found in parallel. The remaining step is to find a good way to connect the forest. Similar approaches that partition the initial dataset using $k$-means clustering also exist (Zhong et al., 2015; Jothi et al., 2018), though this inherently forms clusters based on Euclidean distances. For all of these clustering-based approaches, the number of components $t$ for the initial forest is easy to control since it exactly corresponds to the number of clusters $k$. Smaller $k$ leads to larger clusters, and hence finding an MST of each $P_i$ is harder. However, there are then fewer components to connect, so there is again a trade-off to consider. We remark that it may seem counterintuitive to use a clustering method as a subroutine for finding an MST, since one of the main reasons to compute an MST is to perform clustering. We stress that Strategy 2 uses a cheap and fast clustering method that identifies sets of points that are somewhat close in the metric space, without focusing on whether they are good clusters for a downstream application. These cheaper clusters are then just part of a broader and more sophisticated hierarchical clustering pipeline.

## B. Proofs

We now provided full details for proving the main approximation results in Section 4 for MFC-Approx. Pseudocode is shown in Algorithm 1.

**Lemma 4.1.** Let $P_i$ and $P_j$ be an arbitrary pair of components and let $\beta > 1$. If $\hat{w}_{ij} > \beta w_{ij}^*$, then

$$\hat{w}_{ij} < \frac{\beta}{\beta - 1} \min\{w_\mathcal{X}(T_i), w_\mathcal{X}(T_j)\}. \tag{7}$$

*Proof.* For each $i \in \{1, 2, \ldots, t\}$, we denote the maximum distance between a point in $P_i$ and its component representative $s_i$ by $\alpha_i = \max_{x \in P_i} d(x, s_i)$. Let $x_i^* \in P_i$ and $x_j^* \in P_j$ be points satisfying $d(x_i^*, x_j^*) = d(P_i, P_j) = w_{ij}^*$. We use the

---
**Algorithm 1** MFC-Approx
___

    **Input:** $\mathcal{X} = \{x_1, x_2, \ldots, x_n\}$, components $\mathcal{P} = \{P_1, P_2, \ldots, P_t\}$, component spanning trees $\{T_1, T_2, \ldots, T_t\}$
    **Output:** Spanning tree for implicit metric graph of $\mathcal{X}$
    **for** $i = 1, 2, \ldots t$ **do**
        Select arbitrary component representative $s_i \in P_i$
5:  **end for**
    **for** $i = 1, 2, \ldots t - 1$ **do**
        **for** $j = i + 1, \ldots, t$ **do**
            $w_{i \to j} = \min_{x_i \in P_i} d(x_i, s_j)$               `// closest a` $P_i$ `node comes to` $s_j$
            $w_{j \to i} = \min_{x_j \in P_j} d(x_j, s_i)$               `// closest a` $P_j$ `node comes to` $s_i$
10:      $\hat{w}_{ij} = \min\{w_{i \to j}, w_{j \to i}\}$                    `// set weight for edge` $(v_i, v_j)$
        **end for**
    **end for**
    $\hat{T}_{\mathcal{P}} = \text{OPTIMALMST}(\{\hat{w}_{ij}\}_{i,j \in [t]})$        `// find optimal MST on complete` $t$`-node graph`
    Return spanning tree $\hat{T}$ of $G_{\mathcal{X}}$ by combining $\bigcup_{i=1}^{t} T_i$ with edges from $\hat{T}_{\mathcal{P}}$.

---

(reverse) triangle inequality and the definition of $\alpha_i$ to see that:

$$d(x_i^*, x_j^*) = \min_{x_j \in P_j} d(x_i^*, x_j) \geq \min_{x_j \in P_j} d(s_i, x_j) - d(s_i, x_i^*) \geq \min_{x_j \in P_j} d(s_i, x_j) - \alpha_i = w_{j \to i} - \alpha_i \geq \hat{w}_{ij} - \alpha_i.$$

Similarly we can show that

$$d(x_i^*, x_j^*) = \min_{x_i \in P_i} d(x_i, x_j^*) \geq \min_{x_i \in P_i} d(s_j, x_i) - d(s_j, x_j^*) \geq \min_{x_i \in P_i} d(s_j, x_i) - \alpha_j = w_{i \to j} - \alpha_j \geq \hat{w}_{ij} - \alpha_j.$$

In other words, we have the bound $w_{ij}^* \geq \hat{w}_{ij} - \min\{\alpha_i, \alpha_j\}$. Combining this with the assumption that $\hat{w}_{ij} > \beta w_{ij}^*$ gives

$$\hat{w}_{ij} > \beta w_{ij}^* \geq \beta \hat{w}_{ij} - \beta \min\{\alpha_i, \alpha_j\} \implies \frac{\beta}{\beta - 1} \min\{\alpha_i, \alpha_j\} > \hat{w}_{ij}.$$

The proof follows from the observation that $\alpha_i \leq w_{\mathcal{X}}(T_i)$. To see why, note that there exists some $x \in P_i$ such that $d(x, s_i) = \alpha_i$. Since $T_i$ is a spanning tree of $P_i$, it must contain a path from $s_i$ to $x$ with sum of edge weights at least $\alpha_i$. $\qquad \square$

In addition to Lemma 4.1, our main approximation guarantees rely on two other simple supporting observations. We include a full proof here for completeness. The first amounts to the observation that a tree has arboricity and degeneracy 1. The second supporting observation deals with MSTs in a graph that includes edges of weight zero.

**Observation 1.** If $T = (V, E_T)$ is an undirected tree, there is a way to orient edges in such a way that every node has at most one outgoing edge.

*Proof.* The proof is constructive. Define an iterative algorithm that removes a minimum degree node at each step and deletes all its incident edges. Orient the deleted edges so that they start at the node that was removed. Note that a tree always contains a node of degree 1, and removing such a node leads to another tree with one fewer node. Thus, this procedure will orient edges of the original graph in such a way that each node has at most one outgoing edge. $\qquad \square$

**Observation 2.** Let $w^{(1)} \colon E \to \mathbb{R}^+$ and $w^{(2)} \colon E \to \mathbb{R}^+$ be two nonnegative weight functions for an undirected graph $G = (V, E)$. Assume there is an edge set $Z \subseteq E$ such that

$$w^{(1)}(i, j) = w^{(2)}(i, j) = 0 \text{ for every } (i, j) \in Z.$$

Then there exist spanning trees $M_1$ and $M_2$ for $G$ such that $M_i$ is an MST for $G$ with respect to $w^{(i)}$ for $i \in \{1, 2\}$, and $M_1 \cap Z = M_2 \cap Z$.

*Proof.* The proof is constructive. Recall that Kruskal's algorithm finds an MST by ordering edges by weight (starting with the smallest and breaking ties arbitrarily in the ordering) and then greedily adds each edge a growing spanning tree if and

only if it connects two previously disconnected components. Fix an arbitrary ordering $\sigma_Z$ of edges in $Z$. When applying Kruskal's algorithm to find minimum spanning trees of $G$ with respect to $w^{(1)}$ and $w^{(2)}$, we can choose orderings for these functions that exactly coincide for the first $|Z|$ edges visited. Namely, we place edges in $Z$ first, using the order given by $\sigma_Z$. The first $|Z|$ steps of Kruskal's algorithm will be identical when building MSTs with respect to $w^{(1)}$ and $w^{(2)}$. Thus, if $M_1$ and $M_2$ are the spanning trees obtained for $w^{(1)}$ and $w^{(2)}$ respectively using this approach, we know these trees will include the same set of edges from $Z$ and discard the same set of edges from $Z$, i.e., $M_1 \cap Z = M_2 \cap Z$. □

To aid in proving our main approximation results, we first cover some useful notation. Let $T_{\mathcal{P}}^*$ represent an MST of $G_{\mathcal{P}}$ with respect to the optimal weight function $w^* \colon E_{\mathcal{P}} \to \mathbb{R}^+$ and $\hat{T}_{\mathcal{P}}$ represent an MST of $G_{\mathcal{P}}$ with respect to the approximate weight function $\hat{w} \colon E_{\mathcal{P}} \to \mathbb{R}^+$. The edges of $T_{\mathcal{P}}^*$ map to a set of edges $M^*$ in $G_{\mathcal{X}}$ that optimally solves the METRIC FOREST COMPLETION problem, and the edges in $\hat{T}_{\mathcal{P}}$ map to an edge set $\hat{M}$. The weight of these edges is given by $w_{\mathcal{X}}(M^*) = w^*(T_{\mathcal{P}}^*)$ and $w_{\mathcal{X}}(\hat{M}) = \hat{w}(\hat{T}_{\mathcal{P}})$, respectively. Let $T^*$ be the spanning tree of $G_{\mathcal{X}}$ defined by combining $\bigcup_{i=1}^{t} T_i$ with $M^*$ and $\hat{T}$ be the spanning tree (returned by MFC-Approx) that combines $\bigcup_{i=1}^{t} T_i$ with $\hat{M}$. These have weights given by

$$w_{\mathcal{X}}(T^*) = w^*(T_{\mathcal{P}}^*) + \sum_{i=1}^{t} w_{\mathcal{X}}(T_i) \tag{8}$$

$$w_{\mathcal{X}}(\hat{T}) = \hat{w}(\hat{T}_{\mathcal{P}}) + \sum_{i=1}^{t} w_{\mathcal{X}}(T_i). \tag{9}$$

We now restate and then prove Theorems 4.2 and 4.3 using the new notation above.

**Theorem 4.2** The spanning tree $\hat{T}$ returned by Algorithm 1 satisfies

$$w_{\mathcal{X}}(T^*) \le w_{\mathcal{X}}(\hat{T}) \le \beta w_{\mathcal{X}}(T^*)$$

for $\beta = (3 + \sqrt{5})/2 < 2.62$, where $T^*$ is a solution to the MFC problem. In other words, $T^*$ is a spanning tree of $G_{\mathcal{X}}$ with minimum weight among all spanning trees that complete the initial forest $\bigcup_{i=1}^{t} T_i$.

*Proof.* For our analysis we consider two hypothetical weight functions $w_0^*$ and $\hat{w}_0$ for $G_{\mathcal{P}} = (V_{\mathcal{P}}, E_{\mathcal{P}})$, defined by zeroing out the $\beta$-unbounded edges in $w^*$ and $\hat{w}$:

$$w_0^*(v_i, v_j) = \begin{cases} w_{ij}^* & \text{if } (v_i, v_j) \text{ is } \beta\text{-bounded, i.e., } \hat{w}_{ij} \le \beta w_{ij}^* \\ 0 & \text{otherwise} \end{cases}$$

$$\hat{w}_0(v_i, v_j) = \begin{cases} \hat{w}_{ij} & \text{if } (v_i, v_j) \text{ is } \beta\text{-bounded, i.e., } \hat{w}_{ij} \le \beta w_{ij}^* \\ 0 & \text{otherwise.} \end{cases}$$

By Observation 2, there exist spanning trees $T_0^*$ and $\hat{T}_0$ for $G_{\mathcal{P}}$ that are optimal with respect to $w_0^*$ and $\hat{w}_0$, respectively, which contain the same exact set of $\beta$-unbounded edges. Let $U$ represent this set of $\beta$-unbounded edges in $\hat{T}_0$ and $T_0^*$. Let $B^*$ be the set of $\beta$-bounded edges in $T_0^*$ and $\hat{B}$ be the set of $\beta$-bounded edges in $\hat{T}_0$. Because $\hat{T}_{\mathcal{P}}$ is an MST with respect to $\hat{w}$ we know that:

$$\hat{w}(\hat{T}_{\mathcal{P}}) \le \hat{w}(\hat{T}_0) = \hat{w}(U) + \hat{w}(\hat{B}). \tag{10}$$

We will use this to upper bound the weight of $\hat{T}_{\mathcal{P}}$ in terms of $T^*$. First we claim that

$$\hat{w}(\hat{B}) \le \beta w^*(T_{\mathcal{P}}^*). \tag{11}$$

This follows from the following sequence of inequalities:

$$
\begin{aligned}
\hat{w}(\hat{B}) &= \hat{w}_0(\hat{B}) && \text{since } \hat{w} \text{ and } \hat{w}_0 \text{ coincide on } \beta\text{-bounded edges} \\
&= \hat{w}_0(\hat{B}) + \hat{w}_0(U) && \text{since } \hat{w}_0 \text{ is zero on } \beta\text{-unbounded edges} \\
&= \hat{w}_0(\hat{T}_0) && \text{since } \hat{T}_0 = \hat{B} \cup U \\
&\leq \hat{w}_0(T_0^*) && \text{since } \hat{T}_0 \text{ is optimal for } \hat{w}_0 \\
&= \hat{w}_0(B^*) && \text{since } \hat{w}_0 \text{ is zero on } \beta\text{-unbounded edges} \\
&\leq \beta w_0^*(B^*) && \text{since } \hat{w}_0 \leq \beta w_0^* \text{ on } \beta\text{-bounded edges} \\
&= \beta w_0^*(T_0^*) && \text{since } T_0^* = B^* \cup U \text{ and } w_0^*(U) = 0 \\
&\leq \beta w_0^*(T_\mathcal{P}^*) && \text{since } T_0^* \text{ is optimal for } w_0^* \\
&\leq \beta w^*(T_\mathcal{P}^*) && \text{since } w_0^* \leq w^* \text{ for all edges.}
\end{aligned}
$$

Next we bound $\hat{w}(U)$. From Lemma 4.1, we know that $\hat{w}_{ij} \leq \beta/(\beta-1)\min\{w_\mathcal{X}(T_i), w_\mathcal{X}(T_j)\}$ for every $(v_i, v_j) \in U$. Because $\hat{T}_\mathcal{P}$ is a tree on $G_\mathcal{P}$, we know by Observation 1 that we can orient its edges in such a way that each node in $V_\mathcal{P} = \{v_1, v_2, \ldots, v_t\}$ has at most one outgoing edge. We can therefore assign each $(v_i, v_j) \in U$ to one of its nodes in such a way that each node in $V_\mathcal{P}$ is assigned at most one edge from $U$. Assume without loss of generality that we write edges in such a way that edge $(v_i, v_j) \in U$ is assigned to node $v_i$. Thus,

$$
\hat{w}(U) = \sum_{(v_i, v_j) \in U} \hat{w}_{ij} \leq \sum_{(v_i, v_j) \in U} \frac{\beta}{\beta-1} w_\mathcal{X}(T_i) \leq \frac{\beta}{\beta-1} \sum_{i=1}^{t} w_\mathcal{X}(T_i). \tag{12}
$$

Combining these gives our final bound

$$
\begin{aligned}
w_\mathcal{X}(\hat{T}) &= \hat{w}(\hat{T}_\mathcal{P}) + \sum_{i=1}^{t} w_\mathcal{X}(T_i) && \text{by Eq. (9)} \\[2mm]
&\leq \hat{w}(\hat{B}) + \hat{w}(U) + \sum_{i=1}^{t} w_\mathcal{X}(T_i) && \text{by Eq. (10)} \\[2mm]
&\leq \beta w^*(T_\mathcal{P}^*) + \left( \frac{\beta}{\beta-1} + 1 \right) \sum_{i=1}^{t} w_\mathcal{X}(T_i) && \text{by Eqs. (11) and (12)} \\[2mm]
&\leq \max\left\{ \beta, \frac{\beta}{\beta-1} + 1 \right\} \left( w^*(T_\mathcal{P}^*) + \sum_{i=1}^{k} w_\mathcal{X}(T_i) \right) \\[2mm]
&= \beta w_\mathcal{X}(T^*) && \text{by Eq. (8) and our choice of } \beta
\end{aligned}
$$

For the last step that we have specifically chosen $\beta = (3 + \sqrt{5})/2$ to ensure that $\beta = 1 + \beta/(\beta-1)$, as this leads to the best approximation guarantee using the above inequalities. $\qquad\square$

**Theorem 4.3** Let $G_\mathcal{X}$ be an implicit metric graph and $\mathcal{P}$ be an initial partitioning with $\gamma$-overlap $\gamma = \gamma(\mathcal{P})$. Algorithm 1 returns a spanning tree of $\hat{T}$ of $G_\mathcal{X}$ that satisfies

$$
w_\mathcal{X}(T_\mathcal{X}) \leq w_\mathcal{X}(\hat{T}) \leq \beta w_\mathcal{X}(T_\mathcal{X}) \tag{13}
$$

where $T_\mathcal{X}$ is an MST of $G_\mathcal{X}$ and $\beta = \frac{1}{2}\left(2\gamma + 1 + \sqrt{4\gamma + 1}\right) \leq 2\gamma + 1$.

*Proof.* We use the same terminology and notation as in the proof of Theorem 4.2. The only difference is that we do not necessarily use $\beta = (3 + \sqrt{5})/2$. For an arbitrary $\beta \geq 1$, we can still prove in the same way that

$$
w_\mathcal{X}(\hat{T}) \leq \beta w^*(T_\mathcal{P}^*) + \left( \frac{\beta}{\beta-1} + 1 \right) \sum_{i=1}^{t} w_\mathcal{X}(T_i). \tag{14}
$$

The $\gamma$-overlap of the initial forest implies there exists an MST $T_\mathcal{X}$ of $G_\mathcal{X}$ satisfying:

$$\sum_{i=1}^{t} w_\mathcal{X}(T_i) = \gamma w_\mathcal{X}(I_\mathcal{X}), \tag{15}$$

where $I_\mathcal{X}$ is the set of edges of $T_\mathcal{X}$ inside components $\mathcal{P}$ of the initial forest. Let $B_\mathcal{X}$ be the set of edges in $T_\mathcal{X}$ that cross between components, so that $w_\mathcal{X}(T_\mathcal{X}) = w_\mathcal{X}(B_\mathcal{X}) + w_\mathcal{X}(I_\mathcal{X})$. Since $T_\mathcal{X}$ is a spanning tree, $B_\mathcal{X}$ must contain a path between every pair of components, meaning that $B_\mathcal{X}$ corresponds to a spanning subgraph of the coarsened graph $G_\mathcal{P}$. Since $T_\mathcal{P}^*$ defines an MST of $G_\mathcal{P}$ with respect to $w^*$, which captures the minimum distances between pairs of components, we know

$$w^*(T_\mathcal{P}^*) \le w_\mathcal{X}(B_\mathcal{X}). \tag{16}$$

Putting the pieces together we see that

$$w_\mathcal{X}(\hat{T}) \le \beta w_\mathcal{X}(B_\mathcal{X}) + \left(1 + \frac{\beta}{\beta - 1}\right) \gamma w_\mathcal{X}(I_\mathcal{X}) \le \max\left\{\beta, \gamma\left(1 + \frac{\beta}{\beta - 1}\right)\right\} w_\mathcal{X}(T_\mathcal{X}). \tag{17}$$

This will hold for any choice of $\beta \ge 1$. In order to prove the smallest approximation guarantee, we choose $\beta$ satisfying:

$$\beta = \gamma\left(1 + \frac{\beta}{\beta - 1}\right).$$

The solution for this equation under constraint $\beta \ge 1$ and $\gamma \ge 1$ is

$$\beta = \frac{1}{2}\left(2\gamma + 1 + \sqrt{4\gamma + 1}\right) \le 2\gamma + 1.$$

$\square$

## C. Additional Details for MFC Framework

This section of the appendix provides additional details about our algorithm MFC-Approx and our broader framework for metric MST computation. This includes a more detailed runtime analysis and several practical considerations about computing initial forests.

### C.1. Runtime analysis for MFC-Approx

MFC-Approx finds the distance between each point in $\mathcal{X}$ and each of the $t$ component representatives, for a total of $O(nt)$ distance queries. It then finds an MST of a dense graph with $\binom{t}{2}$ edges, which has runtime and space requirements of $\tilde{O}(t^2)$. Thus, the algorithm has subquadratic memory and query complexity as long as $t = o(n)$. The runtime is $\tilde{O}(nt\mathcal{Q}_\mathcal{X} + t^2)$ where $\mathcal{Q}_\mathcal{X}$ is the complexity for one distance query in $\mathcal{X}$, which also is subquadratic as long as $t\mathcal{Q}_\mathcal{X} = o(n)$. In settings where $\mathcal{Q}_\mathcal{X} = \tilde{O}(1)$, the memory, runtime, and query complexity are all subquadratic as long as $t = o(n)$.

### C.2. Practical considerations for our MFC framework

The practical utility of the MFC framework will also depend on the time it takes to find an initial forest. The precise runtime for computing a such a forest depends on various design choices and trade-offs when using any strategy. To provide some intuition for the overall cost of applying the MFC framework, we provide a rough complexity analysis for the $k$-center strategy (see Appendix A) assuming an idealized case of balanced clusters. The simple 2-approximation for $k$-center chooses an arbitrary first cluster center, and chooses the $i$th cluster center to be the point with maximum distance from the first $i - 1$ centers (Gonzalez, 1985). This requires $O(nt)$ distance queries. For this strategy, we can use the cluster centers as the component representatives for MFC-Approx, which allows us to compute $\hat{w}$ without any additional queries. If clusters are balanced in size, we can compute minimum spanning trees for all clusters using $O(n^2/t)$ queries and a runtime of $\tilde{O}(\mathcal{Q}_\mathcal{X} n^2/t)$, simply by querying all inner-cluster edges and running a standard MST algorithm. In this balanced-cluster case, combining the initial forest complexity with the complexity of our MFC algorithm, the entire pipeline for finding a spanning tree takes $\tilde{O}(\mathcal{Q}_\mathcal{X}(n^2/t + nt) + t^2)$. This is minimized by choosing $t = \sqrt{n}$ clusters, leading to a complexity that grows as $n^{1.5}$. For unbalanced clusters, one must consider different trade-offs for cluster-balancing strategies, which could

be beneficial for runtime but may affect initial cluster quality. We could also improve the runtime at the expense of initial forest quality by not computing an exact MST for each cluster. For example, we could recursively apply our entire MFC framework to find a spanning tree of each cluster. We cover other practical considerations and runtimes for certain ways of computing initial forests in Appendix A.

There are several other ways to relax our overall MFC framework and algorithm to make our approach faster while still satisfying strong approximation guarantees. For metrics with high query complexity, we can use approximate queries with only minor degradation in approximation guarantees. For example, for high-dimensional Euclidean distance we can apply Johnson-Lindenstrauss transformations to reduce the query complexity while approximately maintaining distances. As another relaxation, we can replace the exact nearest neighbor search steps in MFC-Approx (to find the minimum distance between a representative $s_i$ in an entire cluster $P_j$) with an approximate nearest neighbor search. If for some $\varepsilon > 0$ we find a $(1+\varepsilon)$-approximate nearest neighbor in each component for every component representative $s_i$, this will make our approximation guarantees worse by at most a factor $(1+\varepsilon)$. There are also numerous opportunities for parallelization, such as parallelizing distance queries and MST computation for components.

There are also several ways in which we could potentially improve the spanning tree quality of our algorithm in practice with little effect on runtime. As one specific example, when approximating the distance between components $P_i$ and $P_j$ of the initial forest, we could compute $\tilde{x}_i = \operatorname{argmin}_{x \in P_i} d(x, s_j)$ and $\tilde{x}_j = \operatorname{argmin}_{x \in P_j} d(x, s_i)$ and then use the following weight for the coarsened graph:

$$\tilde{w}_{ij} = \min\{d(\tilde{x}_i, s_j), d(\tilde{x}_j, s_i), d(\tilde{x}_j, \tilde{x}_i)\}.$$

This differs from Algorithm 1 only in that it additionally checks the distance $d(\tilde{x}_j, \tilde{x}_i)$ to see if this provides an even closer pair of points between $P_i$ and $P_j$. Although this does not always improve results, it can never be worse in terms of approximations. Figure 3a provides an example where $\tilde{w}_{ij}$ is noticeably better than $\hat{w}_{ij}$. An interesting future direction is to implement this and also explore other heuristics that could improve the practical performance of our method without affecting our theoretical guarantees.

## D. Additional Details on Empirical Results

In this section we provide more in-depth explanations behind our experimental setup, implementations, evaluation, and empirical results.

**Generating initial forests.** To generate initial forests, we first partition data points using the standard greedy 2-approximation algorithm for $k$-center clustering (Gonzalez, 1985), as this is simple, fast, and works for arbitrary metrics. We consider results for a range of different component numbers $t = k$. After partitioning the data, we compute an exact minimum spanning tree for each component, mirroring an approach used by previous divide-and-conquer algorithms for large-scale Euclidean MST approximations (Jothi et al., 2018; Zhong et al., 2015; Mishra & Mohanty, 2020). Computing exact MSTs for all $t$ components is often the most expensive step of our MFC framework as it requires generating all edges inside each component. Even so, we find this leads to significant runtime improvements over applying the exact algorithm to the entire dataset, while achieving extremely good approximation ratios.

**Baseline and evaluation criteria.** We compare against an exact MST computed by generating all $\binom{n}{2}$ edge weights then running Kruskal's algorithm. The bottleneck in Kruskal's algorithm is sorting the $O(n^2)$ edge weights, leading to an $O(n^2 \log n)$ runtime. Although more sophisticated algorithms could achieve a runtime of $O(n^2)$ for arbitrary metrics, Kruskal's algorithm is still fast (optimal up to a log factor in terms of runtime) and has the added benefit of being simple to implement and work with in practice. We note furthermore that any speed-up in our baseline algorithm for the full MST would also instantly improve the runtime of our MFC framework, since we use the same solver to find MSTs for the partitions in the initial forest.

We evaluate our MFC framework (both the initial forest generation and the completion step) in terms of runtime, a bound on $\gamma$-overlap, and cost ratio (i.e., the spanning tree cost divided by the weight of the optimal MST). To find our upper bound for $\gamma$, which we denote as $\bar{\gamma}$, we compute the overlap between the initial forest and the MST found by the baseline. This amounts to computing the ratio in Eq. (3), except without optimizing over all possible minimum spanning trees in the denominator. When all edge weights are unique, this produces the exact value of $\gamma$. We therefore expect this to be an especially good approximation of $\gamma$ for our Euclidean point cloud datasets, since distances between random points tend to be unique (or nearly all unique).

**Real-world datasets.** We run experiments on several real-world datasets that have been used as benchmarks for similarity search algorithms (e.g., $k$-NN search benchmarks) and clustering algorithms. These serve well as benchmarks for metric MST computations since they involve a variety of different types of data (e.g., numerical data, sets, strings) and distance functions (e.g., Euclidean, Jaccard, Hamming, Levenshtein edit distance). In some cases we used the entire dataset in our numerical experiments, whereas in other cases we subsample a larger dataset. We run experiments on implicit graphs with up to $n = 39{,}774$ nodes, which means we are considering graphs with over 700 million edges. Our MFC framework in fact scales up to even larger graphs, but we restrict to graphs with under 1 billion edges since in this regime we are still able to compute an optimal MST to use as a point of comparison. We summarize the dataset type and distance metric in Table 2, and provide more details for each dataset below.

- **Cooking.** Each data object is a set of food ingredients defining a recipe. There are 6714 ingredients and 39,774 recipes. We use Jaccard distance as this is set data. The original raw data comes from the *What's Cooking?* Kaggle dataset (Kaggle, 2015), which was parsed into the current set of sets by Amburg et al. (2020).

- **MovieLens-10M.** This is a set of sets derived from movie ratings (Harper & Konstan, 2015). We consider a subset of the parsed dataset provided on the ANN Benchmarks Repository (Aumueller et al., 2024), restricting to sets with 64 items or more, in order to work with a dataset where $n$ is slightly larger than 30,000. We apply Jaccard distance.

- **Kosarak.** This dataset is derived from click-stream data from a Hungarian news portal (Bodon) and is a standard benchmark for nearest neighbor algorithms (Aumueller et al., 2024). The original data (available from the Frequent Itemset Mining Dataset Repository at `http://fimi.uantwerpen.be/data/kosarak.dat`) comprises 990,002 sets defined over a collection of 41,270 items. We restricted to sets of size at least 40, leading to a set of 32,295 sets. We apply Jaccard distance.

- **Fashion-MNIST.** Each point in the Fashion-MNIST dataset represents a $28 \times 28$ grayscale image of an item of clothing (e.g., sneakers, trousers), encoded as a vector of size $d = 784$. The total number of class labels is 10. We use Euclidean distance for this dataset, following the ANN Benchmarks Repository (Aumueller et al., 2024).

- **Names-US.** Each data object is a last name for someone in the United States, obtained from the Names dataset available online at `https://github.com/philipperemy/name-dataset`. We consider this dataset as it provides a natural benchmark for computing spanning trees on short sequence data using Levenshtein edit distance. Each UTF-8 code point was treated as a separate character for the purpose of the edit distance. Last names in the dataset vary in length from 0 to 252 characters where 0 indicates no last name. The average length of the last names is 6.67 with a standard deviation of 2.470. Computing an exact minimum spanning tree for the entire (very large) dataset is infeasible, so our experimental results consider samples of names of size $n = 30{,}000$.

- **GreenGenes-Unaligned and GreenGenes-Aligned.** GreenGenes is a chimera-checked 16S megagenomic sequence dataset (DeSantis et al., 2006), which is frequently used as a benchmark for sequence clustering. We use version 13.5 of the data, accessed by following instructions on the USEARCH benchmarks website (`https://www.drive5.com/usearch/benchmark_ggclust.html`). The data is provided in two formats: as unaligned variable length sequences (ranging from 1111 to 2368 characters; mean length is 1401.06 with a standard deviation of 57.083), and as aligned sequences of a fixed length of 7682. Following a standard and natural approach, we used Levenshtein edit distance for the unaligned sequences and Hamming distance for the aligned sequences. Since the dataset is far too large to compute a full spanning tree, we subsample groups of $n$ sequences at a time. Computing Levenshtein distances on unaligned sequences is very expensive, so we restrict to $n = 2500$ for this case.

### D.1. Additional details for uniform random data

For our uniform random data experiments we consider random point clouds of size $n$ in $d$-dimensional Euclidean space where the value for a point in each dimension is drawn uniformly from $[-1, 1]$. The first two columns of Figure 4 show results for $d \in \{8, 256\}$ as $n$ increases from 1000 to 30000, using a different color for each choice of component number $t \in \{16, 32, 64, 128, 256\}$ for which we ran our MFC framework. Here in Figure 7 we show results for for $d \in \{4, 16, 64, 128\}$, which show the same basic trends.

As $t$ increases, we see larger and larger speedups over the exact baseline algorithm, with up to a 300x speedup for $d = 8$ and up to a 30x speedup for $d = 256$. Our results match the asymptotic behavior we would expect from a simple runtime

Table 2: List of different datasets used for real world experiments.

| Dataset | Type of Data | Distance Metric |
|---|---|---|
| Cooking (Amburg et al., 2020) | Sets | Jaccard distance |
| Kosarak (Bodon) | Sets | Jaccard distance |
| MovieLens-10M (Harper & Konstan, 2015) | Sets | Jaccard distance |
| Fashion-MNIST (Xiao et al., 2017) | 784 dimensional points | Euclidean distance |
| Names-US (Remy, 2021) | Short variable-length strings | Levenshtein edit distance |
| GreenGenes-Unaligned (DeSantis et al., 2006) | Long variable-length strings | Levenshtein edit distance |
| GreenGenes-Aligned (DeSantis et al., 2006) | 7682 character strings | Hamming distance |

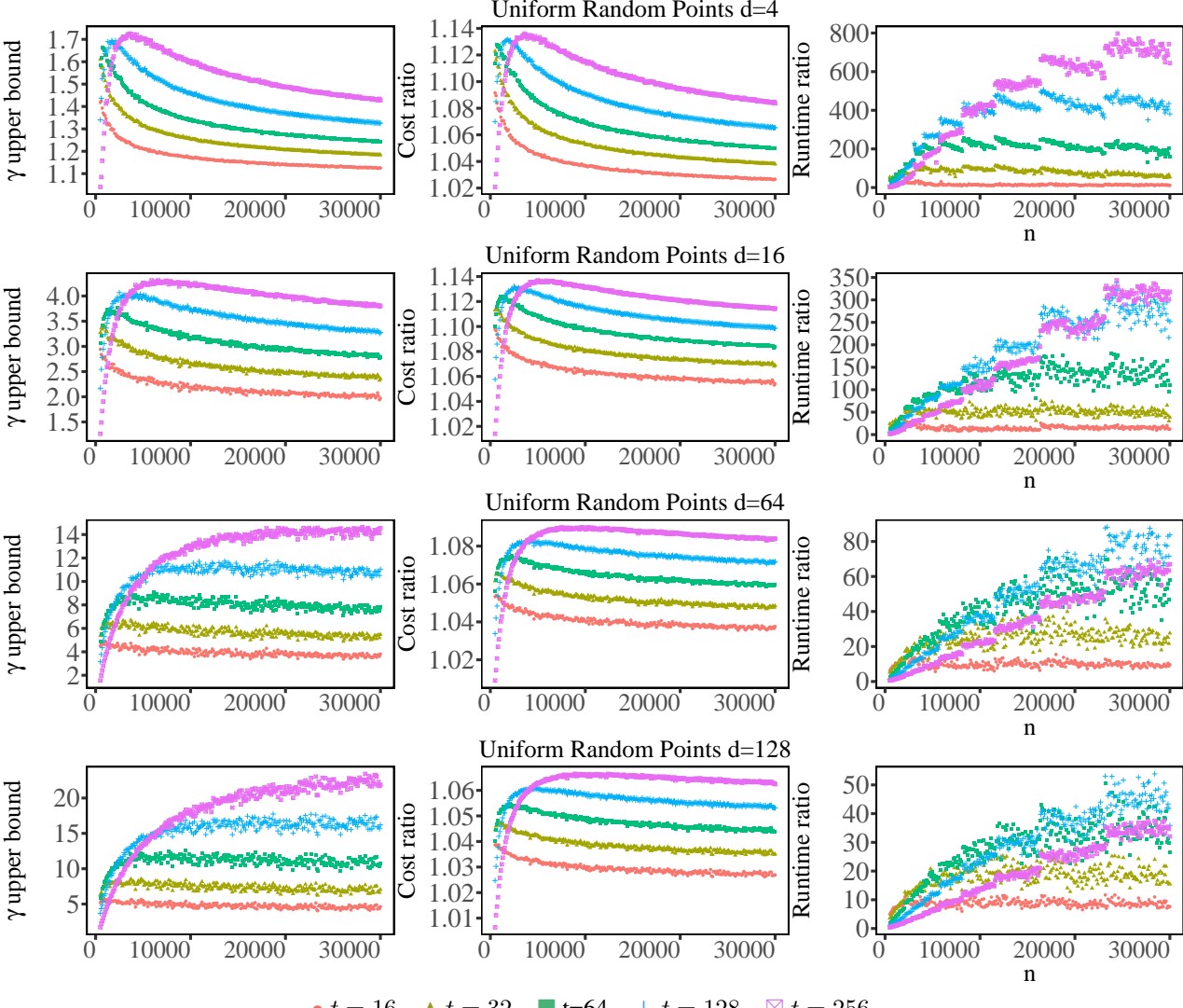

Figure 7: Results on synthetic uniform random data for dimensions $d \in \{4, 16, 64, 128\}$. Each point in each plot represents an average over 16 sampled point clouds for a fixed $n$ and choice of component number $t$. Runtime ratio is the ratio between the runtime for the optimal MST algorithm divided by the runtime of our MFC framework (including initial forest generation). Cost ratio is the ratio between the spanning tree weight for our method and the optimal MST weight. The $\gamma$ upper bound is computed by comparing the initial forest overlap with the one optimal MST computed.

analysis: for a fixed value of $t$ the ratio between the exact algorithm runtime and the runtime of the MFC framework converges to a constant as $n$ increases. As $t$ or $d$ increases, we need a larger value of $n$ to reach this asymptotic speedup. As $t$ increases and the asymptotic runtime improves, the quality of the approximate spanning tree decreases slightly, but the cost approximation ratio still remains very good (i.e., close to 1) in all cases. The $\gamma$-overlap bound $\bar{\gamma}$ tend to be small for low dimensions, but gets large as $d$ increases (e.g., $\bar{\gamma}$ close to 30 for $d = 256$). Nevertheless, cost ratios remain very close to 1, and even improve slightly as $d$ increases.

### D.2. Additional details for clustered data

For our experiments on Gaussian synthetic data (Figure 6), we generate a mixture of $g$ Gaussians in $d$-dimensional Euclidean space with $g$ ranging from 8 to 300. For each Gaussian we generate $\lfloor 20000/g \rfloor$ points, so that $n \approx 20000$ for each dataset. The mean of each Gaussian is chosen uniformly at random from an 8-dimensional box where each axis ranges from $-5$ to $5$. The standard deviation of each dimension of each Gaussian is chosen uniformly at random between $0.5$ and $0.8$. In practice these parameters generate good (but not perfect) clustering structure. We then run our MFC framework for each choice of initial component number $t \in \{16, 32, 64, 128, 256\}$.

Figure 8 shows results for a wider range of $d$ values and includes runtime ratio plots as well. We see good performance in runtime improvements, with larger values of $t$ leading to a larger asymptotic speedup (up to a 400x speedup for $t = 256$ and $d = 8$). The cost ratio and $\bar{\gamma}$ values show the same basic trend for all dimensions: the quality of the spanning tree returned by our method is the best when $t \approx g$. Thus, when we have a good estimate of the number of number of true underlying clusters in a dataset, we can leverage this information when generating an initial forest and running our MFC framework.

### D.3. Additional details and results for real-world data

In Table 3 we include expanded results for the real-world datasets. This includes results for a wider range of $t$ values (number of components generated for the initial forest), and details for the proportion of time spent on the three main steps of our MFC framework:

1. $k$-**centering**: obtaining the initial partitioning $\mathcal{P}$ using a greedy 2-approximaton for $k$-centering (Gonzalez, 1985).

2. **Sub-MST**: Using Kruskal's algorithm to find optimal MSTs $\{T_i\}_{i=1}^{t}$ for the components of $\mathcal{P}$.

3. **MFC-Approx**: Approximating the MFC problem using MFC-Approx.

As anticipated, the most expensive step in this pipeline is typically computing optimal MSTs for the components (Sub-MST step). However, as $t$ increases, the $k$-centering step and the MFC-Approx algorithm become a significant portion of the overall runtime, and even dominate in extreme cases. We remark that the MFC-Approx step is always dominated by the time it takes to form the coarsened graph $G_{\mathcal{P}}$. The time spent finding an MST of $G_{\mathcal{P}}$ is negligible for all datasets and choices of $t$ we considered.

We note several trends in the performance of our MFC framework from Table 3. Across all datasets (and corresponding distance metrics), the $\bar{\gamma}$ values tend to be very good: usually below 2 and never above 3.2. As expected, $\bar{\gamma}$ values get slightly worse as $t$ increases. This is also true for cost ratios, but cost ratios nevertheless remain very good in all cases (always below 1.2 and typically much better). In general, runtimes improve as $t$ increases up to a certain point. Once $t$ becomes too large, computing MSTs for the components found by $k$-centering is no longer the bottleneck. When the number of components becomes too large, the $k$-centering step and building the coarsened graph become too expensive. This is especially clear on the GreenGenes-Unaligned dataset, but the same behavior can be observed in other cases as well.

**Names-US Dataset** One exception to the general runtime trends is that running the MFC framework on the Names-US dataset, for small values of $t$, is in fact slower than computing an optimal MST. On this dataset, we found that the $k$-centering step in our MFC framework tends to form a disproportionately large component, and finding an MST of a very large component takes nearly as long as finding an MST of the entire dataset. Combining this with the time taken on other steps in our MFC framework can lead to a larger overall runtime. This behavior (having a single large cluster) is likely tied to the fact that the Names-US dataset contains extreme outlier strings which causes our simple $k$-centering implementation to create tiny clusters for these outliers, while grouping most other non-outlier strings into a single large cluster. Despite this bad behavior, we note that runtime ratios do continue to increase as $t$ increases and our MFC framework is faster for the largest $t$ values.

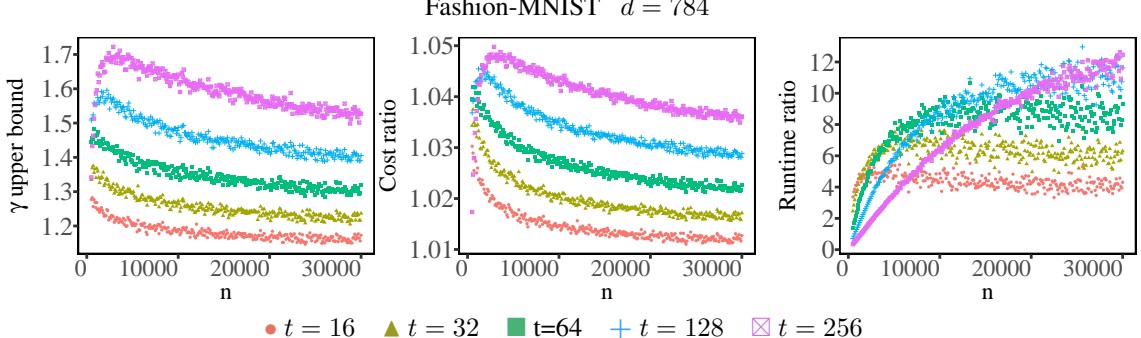

Figure 8: Results on synthetic data with clustering structure: mixtures of $g$ Gaussians for various $d$. Cost ratio and $\gamma$-overlap bound $\bar{\gamma}$ is typically minimized when the number of Gaussians $g$ roughly matches the number of components $t$ used by our MFC framework.

Figure 9: Results for Fashion-MNIST. Each point is the average of 16 samples for fixed $n$ and $t$.

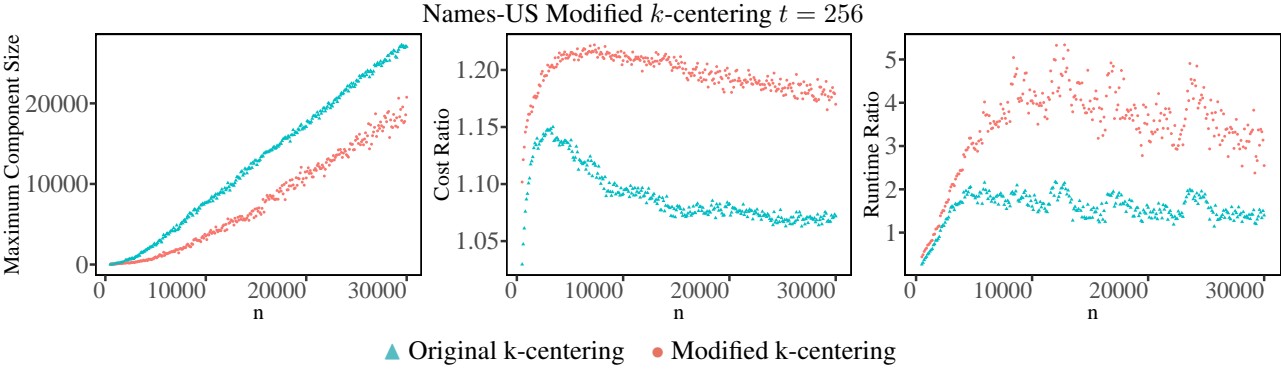

Figure 10: Results comparing a slightly modified version of $k$-centering to the unmodified version on the Names-US dataset when $t = 256$. The $x$-axis reports the number of names $n$ sampled from the larger dataset. The modified version of $k$-centering first runs normal $k$-centering with $k = t/2$, then it further refines the largest cluster by running $k$-centering on that cluster with $k = t/2 + 1$, resulting in a total of $t$ components. Additionally, when the distance between a point and multiple cluster centers is equal, the point is added to the smallest of those clusters. This reduces the size of the largest component (left plot), resulting in slightly worse cost ratios (middle plot) but better runtime improvements (right plot).

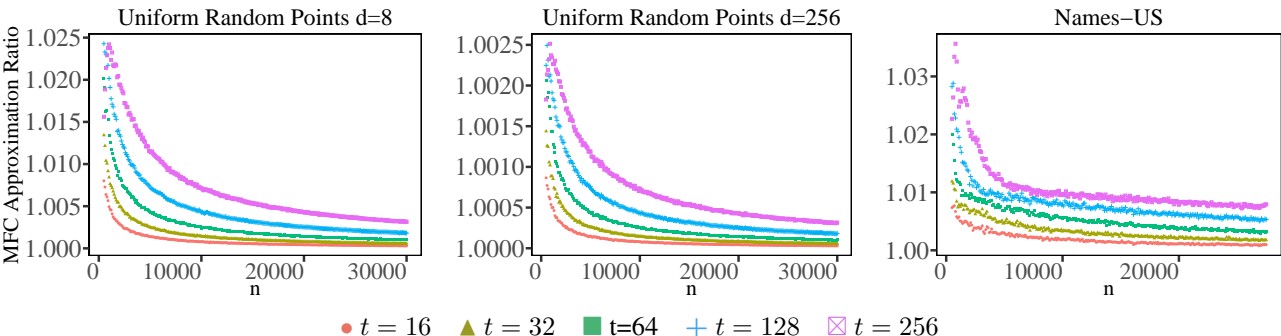

Figure 11: Results showing the MFC approximation ratio achieved on both synthetic and real world data. The MFC approximation ratio equals the weight of the spanning tree produced using our MFC approximation divided by the weight of the spanning tree produced by optimally completing the initial forest (i.e., optimally solving MFC). The input to the MFC algorithm here is produced using the same process as previous experiments, using $k$-centering for clustering and optimally solving the MST problem on individual components.

We found that a slightly modified version of our $k$-centering step provides a simple way to improve results on the Names-US dataset. This modified version first runs normal $k$-centering to form $k = t/2$ initial clusters, then runs the $k$-centering algorithm again on the largest cluster to break it into $t/2 + 1$ components. This creates the same number of components as running $k$-centering once with parameter $k = t$ on the dataset. Additionally, when iterating through points to assign them to clusters after the centers have been chosen, if a point has the same distance to multiple cluster centers, this modified $k$-centering algorithm assigns the point to the cluster with the current fewest number of points. Figure 10 shows results for the full MFC pipeline on the Names-US dataset when we used this modified version of $k$-centering for $t = 256$. We also show results in a new entry at the end of Table 3. We can see that this simple modification to the $k$-centering algorithm improves the results significantly on the Names-US dataset for both small and large values of $t$.

## D.4. Approximation for MFC Problem

Our previous results all focused on approximations for the MST problem using our MFC framework, i.e., comparing in-practice approximations against the bound in Theorem 4.3 given in terms of $\gamma$. Figure 11 shows approximation ratios for just the METRIC FOREST COMPLETION problem. In other words, we compare against the weight of the best spanning tree that completes the initial forest, rather than against the weight of an optimal spanning tree for the original dataset. Theorem 4.2 shows this ratio must be less than 2.62. In practice, we are consistently under 1.1, far better than the theoretical guarantee.

Table 3: Results for real-world datasets using various types of metrics. Runtimes are in minutes. For the last 5 datasets we report averages and standard deviations for 16 different samples of size $n$. The last three rows for each dataset report the proportion of time spent on each step of the MFC framework.

| Dataset | | OPT | t = 16 | t = 32 | t = 64 | t = 128 | t = 256 |
|---|---|---|---|---|---|---|---|
| Cooking | $\bar{\gamma}$ | - | 1.770 | 2.014 | 2.222 | 2.421 | 3.144 |
| | Cost Ratio | 1 | 1.040 | 1.051 | 1.059 | 1.069 | 1.089 |
| $n = 39774$ | Runtime Ratio | 1 | 4.391 | 6.524 | 8.689 | 9.966 | 15.598 |
| | Runtime (mins) | $24.2_{\pm 0.61}$ | 5.3 | 3.5 | 2.7 | 2.3 | 1.5 |
| | k-centering % | - | 0.006 | 0.016 | 0.038 | 0.082 | 0.321 |
| | Sub-MST % | - | 0.989 | 0.971 | 0.923 | 0.826 | 0.355 |
| | MFC-approx % | - | 0.004 | 0.013 | 0.038 | 0.092 | 0.324 |
| MovieLens | $\bar{\gamma}$ | - | 1.395 | 1.513 | 1.690 | 1.955 | 2.283 |
| | Cost Ratio | 1 | 1.010 | 1.012 | 1.017 | 1.022 | 1.028 |
| $n = 33240$ | Runtime Ratio | 1 | 4.026 | 5.014 | 6.287 | 9.337 | 9.529 |
| | Runtime (mins) | $536.0_{\pm 15.58}$ | 120.3 | 96.6 | 77.0 | 51.9 | 50.8 |
| | k-centering % | - | 0.016 | 0.028 | 0.054 | 0.130 | 0.238 |
| | Sub-MST % | - | 0.978 | 0.959 | 0.912 | 0.765 | 0.508 |
| | MFC-approx % | - | 0.006 | 0.013 | 0.034 | 0.104 | 0.253 |
| Kosarak | $\bar{\gamma}$ | - | 1.289 | 1.737 | 2.281 | 2.788 | 3.129 |
| | Cost Ratio | 1 | 1.007 | 1.013 | 1.020 | 1.025 | 1.029 |
| $n = 32295$ | Runtime Ratio | 1 | 2.682 | 5.994 | 15.159 | 18.174 | 12.583 |
| | Runtime (mins) | $182.2_{\pm 0.26}$ | 68.0 | 30.4 | 12.0 | 10.0 | 14.5 |
| | k-centering % | - | 0.010 | 0.036 | 0.146 | 0.363 | 0.464 |
| | Sub-MST % | - | 0.985 | 0.935 | 0.725 | 0.339 | 0.138 |
| | MFC-approx % | - | 0.005 | 0.028 | 0.128 | 0.297 | 0.397 |
| GreenGenes-Unalign. | $\bar{\gamma}$ | - | $1.388_{\pm 0.08}$ | $1.460_{\pm 0.07}$ | $1.460_{\pm 0.06}$ | $1.402_{\pm 0.04}$ | $1.314_{\pm 0.02}$ |
| | Cost Ratio | 1 | $1.092_{\pm 0.02}$ | $1.104_{\pm 0.01}$ | $1.095_{\pm 0.01}$ | $1.075_{\pm 0.01}$ | $1.055_{\pm 0.00}$ |
| $n = 2500$ | Runtime Ratio | 1 | $2.555_{\pm 0.55}$ | $3.270_{\pm 0.52}$ | $3.156_{\pm 0.36}$ | $2.116_{\pm 0.08}$ | $1.159_{\pm 0.04}$ |
| | Runtime (mins) | $239.3_{\pm 6.24}$ | $98.0_{\pm 23.0}$ | $74.9_{\pm 11.9}$ | $76.7_{\pm 8.8}$ | $113.2_{\pm 4.0}$ | $206.5_{\pm 6.0}$ |
| | k-centering % | - | $0.068_{\pm 0.01}$ | $0.172_{\pm 0.03}$ | $0.331_{\pm 0.03}$ | $0.448_{\pm 0.01}$ | $0.492_{\pm 0.01}$ |
| | Sub-MST % | - | $0.885_{\pm 0.03}$ | $0.692_{\pm 0.06}$ | $0.377_{\pm 0.08}$ | $0.134_{\pm 0.01}$ | $0.047_{\pm 0.01}$ |
| | MFC-approx % | - | $0.047_{\pm 0.01}$ | $0.136_{\pm 0.03}$ | $0.292_{\pm 0.05}$ | $0.418_{\pm 0.01}$ | $0.461_{\pm 0.01}$ |
| GreenGenes-Aligned | $\bar{\gamma}$ | - | $1.224_{\pm 0.07}$ | $1.368_{\pm 0.10}$ | $1.466_{\pm 0.06}$ | $1.512_{\pm 0.05}$ | $1.531_{\pm 0.03}$ |
| | Cost Ratio | 1 | $1.074_{\pm 0.02}$ | $1.117_{\pm 0.03}$ | $1.143_{\pm 0.02}$ | $1.147_{\pm 0.01}$ | $1.141_{\pm 0.01}$ |
| $n = 30000$ | Runtime Ratio | 1 | $1.604_{\pm 0.51}$ | $2.375_{\pm 0.82}$ | $3.918_{\pm 1.99}$ | $5.849_{\pm 2.83}$ | $7.026_{\pm 1.69}$ |
| | Runtime (mins) | $33.1_{\pm 0.17}$ | $22.1_{\pm 4.9}$ | $15.1_{\pm 3.7}$ | $9.9_{\pm 3.1}$ | $6.7_{\pm 2.6}$ | $5.0_{\pm 1.3}$ |
| | k-centering % | - | $0.003_{\pm 0.00}$ | $0.010_{\pm 0.00}$ | $0.033_{\pm 0.02}$ | $0.098_{\pm 0.05}$ | $0.232_{\pm 0.05}$ |
| | Sub-MST % | - | $0.994_{\pm 0.00}$ | $0.983_{\pm 0.01}$ | $0.940_{\pm 0.03}$ | $0.817_{\pm 0.09}$ | $0.550_{\pm 0.12}$ |
| | MFC-approx % | - | $0.002_{\pm 0.00}$ | $0.007_{\pm 0.00}$ | $0.027_{\pm 0.02}$ | $0.084_{\pm 0.05}$ | $0.216_{\pm 0.07}$ |
| Fashion-MNIST | $\bar{\gamma}$ | - | $1.173_{\pm 0.02}$ | $1.237_{\pm 0.03}$ | $1.320_{\pm 0.05}$ | $1.405_{\pm 0.03}$ | $1.527_{\pm 0.05}$ |
| | Cost Ratio | 1 | $1.013_{\pm 0.00}$ | $1.017_{\pm 0.00}$ | $1.023_{\pm 0.00}$ | $1.029_{\pm 0.00}$ | $1.036_{\pm 0.00}$ |
| $n = 30000$ | Runtime Ratio | 1 | $4.675_{\pm 1.42}$ | $7.176_{\pm 2.55}$ | $10.027_{\pm 2.85}$ | $12.129_{\pm 2.64}$ | $12.698_{\pm 1.83}$ |
| | Runtime (mins) | $11.4_{\pm 0.31}$ | $2.7_{\pm 0.8}$ | $1.8_{\pm 0.7}$ | $1.2_{\pm 0.4}$ | $1.0_{\pm 0.2}$ | $0.9_{\pm 0.1}$ |
| | k-centering % | - | $0.008_{\pm 0.00}$ | $0.024_{\pm 0.01}$ | $0.067_{\pm 0.02}$ | $0.162_{\pm 0.03}$ | $0.339_{\pm 0.05}$ |
| | Sub-MST % | - | $0.986_{\pm 0.00}$ | $0.963_{\pm 0.01}$ | $0.896_{\pm 0.03}$ | $0.746_{\pm 0.06}$ | $0.462_{\pm 0.07}$ |
| | MFC-approx % | - | $0.004_{\pm 0.00}$ | $0.013_{\pm 0.01}$ | $0.036_{\pm 0.01}$ | $0.090_{\pm 0.02}$ | $0.197_{\pm 0.03}$ |
| Names-US | $\bar{\gamma}$ | - | $1.020_{\pm 0.02}$ | $1.059_{\pm 0.02}$ | $1.139_{\pm 0.04}$ | $1.219_{\pm 0.05}$ | $1.322_{\pm 0.07}$ |
| | Cost Ratio | 1 | $1.005_{\pm 0.00}$ | $1.015_{\pm 0.01}$ | $1.034_{\pm 0.01}$ | $1.051_{\pm 0.01}$ | $1.071_{\pm 0.01}$ |
| $n = 30000$ | Runtime Ratio | 1 | $0.923_{\pm 0.16}$ | $0.954_{\pm 0.18}$ | $0.939_{\pm 0.17}$ | $1.027_{\pm 0.19}$ | $1.138_{\pm 0.21}$ |
| | Runtime (mins) | $2.0_{\pm 0.28}$ | $2.1_{\pm 0.2}$ | $2.1_{\pm 0.2}$ | $2.1_{\pm 0.2}$ | $1.9_{\pm 0.2}$ | $1.7_{\pm 0.1}$ |
| | k-centering % | - | $0.003_{\pm 0.00}$ | $0.005_{\pm 0.00}$ | $0.009_{\pm 0.00}$ | $0.019_{\pm 0.00}$ | $0.038_{\pm 0.00}$ |
| | Sub-MST % | - | $0.995_{\pm 0.00}$ | $0.992_{\pm 0.00}$ | $0.985_{\pm 0.00}$ | $0.970_{\pm 0.01}$ | $0.937_{\pm 0.01}$ |
| | MFC-approx % | - | $0.001_{\pm 0.00}$ | $0.003_{\pm 0.00}$ | $0.005_{\pm 0.00}$ | $0.011_{\pm 0.00}$ | $0.024_{\pm 0.01}$ |
| Names-US | $\gamma$ | - | $1.034_{\pm 0.02}$ | $1.100_{\pm 0.03}$ | $1.224_{\pm 0.04}$ | $1.479_{\pm 0.16}$ | $2.080_{\pm 0.22}$ |
| *Reclustered* | Cost Ratio | 1 | $1.008_{\pm 0.00}$ | $1.026_{\pm 0.01}$ | $1.054_{\pm 0.01}$ | $1.100_{\pm 0.02}$ | $1.170_{\pm 0.02}$ |
| $N = 30000$ | Runtime Ratio | 1 | $1.074_{\pm 0.19}$ | $1.114_{\pm 0.19}$ | $1.246_{\pm 0.25}$ | $1.389_{\pm 0.36}$ | $2.688_{\pm 0.81}$ |
| | Runtime (mins) | $1.9_{\pm 0.31}$ | $1.8_{\pm 0.1}$ | $1.7_{\pm 0.1}$ | $1.5_{\pm 0.1}$ | $1.4_{\pm 0.2}$ | $0.7_{\pm 0.2}$ |
| | k-centering % | - | $0.003_{\pm 0.00}$ | $0.006_{\pm 0.00}$ | $0.012_{\pm 0.00}$ | $0.024_{\pm 0.00}$ | $0.083_{\pm 0.02}$ |
| | Sub-MST % | - | $0.994_{\pm 0.00}$ | $0.990_{\pm 0.00}$ | $0.981_{\pm 0.00}$ | $0.961_{\pm 0.01}$ | $0.860_{\pm 0.04}$ |
| | MFC-approx % | - | $0.002_{\pm 0.00}$ | $0.003_{\pm 0.00}$ | $0.007_{\pm 0.00}$ | $0.015_{\pm 0.01}$ | $0.057_{\pm 0.02}$ |

