# OpenReview forum: "Approximate Forest Completion and Learning-Augmented Algorithms for Metric Minimum Spanning Trees"
_ICML.cc/2025/Conference — ICML 2025 poster_

### Official Review · Reviewer_ki5g · 2025-03-05

**Overall Recommendation:** 4

**Summary:**

The paper considers Metric Forest Completion (MFC) problem.
Let $G$ be an edge-weighted complete graph whose weights satisfy the triangle inequality (it induces a metric space).
Given a forest $F \subseteq G$, the MFC asks for the set of edges with a minimum sum of weights that connects the components of $F$.
They prove that  $\Omega(n^2)$ query edges are required to optimally solve MFC  and present a subquadratic algorithm with an approximation factor $2.62$. In addition, they report an experimental study that shows that their approach provides gains in terms of  scalability without significantly compromising the quality of the produced solutions (spanning trees)

**Claims And Evidence:**

I think they are clear

**Essential References Not Discussed:**

I'm not aware of essential related work that was not cited in the paper.

**Experimental Designs Or Analyses:**

Yes, I checked the experimental designs and analyses and found no major issues.

That said,  I think the behavior of $t$ needs some additional explanation. The time complexity (line 315, left) increases with $t$  but in the experimental results (line 317-319, Right), the speed up increases for larger $t$, which is not in line with the theoretical analysis.
I believe the reason is that in the time complexity analysis, one does not take into account the construction of the initial forest, while in the running times reported in Section 5, it is taken.  In any event, I think it should be clarified. Maybe adding an extra parameter in the time complexity analysis to represent the time taken by the initial heuristic should be helpful.

**Methods And Evaluation Criteria:**

Yes, they make sense

**Other Comments Or Suggestions:**

Some points:

* For Theorem 4.2, I understand you are assuming that $\gamma=1$. Is this correct? In the positive case, you shoud state that.

* line (329, right). I would avoid in "practice" since I do not see  uniform random data as a typical case that occur in practical applications.

**Other Strengths And Weaknesses:**

My positive score is based on

Strengths

* The design of scalable algorithms for MST, in metric space, with provable guarantees is a relevant topic  As the authors mention, and I agree,  for many tasks (some in the Machine Learning domain) the construction of an MST is an important step.

* The paper is, in general, very well written.

*  The proposed framework is simple enough so that one would not have problems reproducing it.

*  Both the improvement in terms of time complexity and the approximation bounds are interesting enough.

* The experiments were well designed


Weakness

* Lack of comparison with other available methods to build MST for metric spaces (even in Euclidean space). I think the paper can be accepted without that, but adding these comparisons would make the paper stronger and more informative regarding what should be used in practice.

**Questions For Authors:**

No relevant questions

**Relation To Broader Scientific Literature:**

Most of the algorithms available for building MST in metric spaces (lines 29-33 and 119-121, right) are only suitable for Euclidean distance.
This work differs from previous work since it handles general metric spaces that include, for instance, edit distance.

For this more general setting, there are lower bounds (Indyk 99) which show that any algorithm with constant approximation for building an MST needs to query/know   $n^2$ edges. To overcome this obstacle (prove theoretical bounds), the algorithm takes the approach of the (recent) field of learning-augmented algorithms,  assuming the availability of a prediction/warm start, obtained via a heuristic or a prediction method. Under this assumption, they obtain an algorithm with an approximation factor that depends on how good the prediction is (parameter $\gamma$).

**Theoretical Claims:**

I have not checked them.

---

> ### Author Rebuttal · Authors · 2025-03-29
>
> Thank you for your feedback on our manuscript!
>
> Thanks for your question about the parameter t, which we agree can be better clarified. As you noticed, the apparent discrepancy comes from whether or not one incorporates the time taken to form the initial forest. The theoretical analysis in lines 315-319 (left column) focuses on our algorithm for the MFC problem specifically, which assumes the initial forest is given. However, because computing initial forests is an important consideration in practice, our runtimes in the experiments include the time for computing an initial forest. We originally wanted to include a full runtime analysis that included the initial forest computation (using the k-centering approach), but were space constrained (hence the reference to Appendix D in line 319 left column). In the appendix, we also include runtimes in Table 3 for each step of the pipeline individually, but again for space constraints we were unable to include this in the main text. Again though, we agree that this could be confusing and we will update the text to better clarify this point.
>
> Regarding algorithm comparisons, we agree that comparing our approach against other heuristics could be interesting. The main issue, as you highlight, is that existing methods are focused only on Euclidean metrics, and we are focused on algorithms that apply to general metric spaces and have theoretical guarantees. With this in mind, we felt that the best experiments section for this paper would be to provide an extensive comparison between our approach and the optimal quadratic algorithm for arbitrary metric spaces, across a wide range of datasets and metrics.
>
> Regarding your question about Theorem 4.2, note that this is a result purely about an approximation for the MFC problem, not the MST problem on the original dataset. As such there is no defined $\gamma$ parameter here since it is a result independent of the overlap between the initial forest and some optimal MST. However, if we did assume that $\gamma = 1$ as you suggest, note that the optimal solution for MFC coincides with the optimal solution to the MST problem on the full dataset. Hence a corollary of this theorem is that if $\gamma = 1$ for the initial forest, we get a 2.62 approximation for the MST problem (not just the MFC problem).
>
> Thanks for your point about the use of “in practice”. We agree and will update this.

---

> > ### Comment · Reviewer_ki5g · 2025-04-02
> >
> > Thank you for the answers!
> > I'm keeping my positive score

---

### Official Review · Reviewer_C5ud · 2025-03-14

**Overall Recommendation:** 4

**Summary:**

The paper presents a new framework for computing approximate minimum spanning trees (MSTs) in arbitrary metric spaces, improving the time complexity compared to traditional exact MST algorithms. The exact MST algorithms require $\tilde{O}(m)$ time complexity, where $m$ is the number of edges. For a complete graph over $n$ vertices, it requires $O(n^2)$ time.

The authors propose the Metric Forest Completion (MFC) problem, which assumes access to an initial forest obtained via heuristics or exact algorithms, and aims to find a set of edges to complete it into a spanning tree. They show that optimally solving MFC still requires $\Omega(n^2)$ queries in the worst case, but they provide a 2.62-approximation algorithm in subquadratic time as long as the number of components $t = o(n)$. Additionally, in the learning-augmented setting, they prove that if the initial forest has a sufficient overlap with an optimal MST, the approximation factor improves based on the overlap parameter $\gamma$. Experimental results on synthetic and real-world datasets demonstrate that the proposed approach finds near-optimal spanning trees significantly faster than exact methods.

**Claims And Evidence:**

The claims are well-supported by theoretical proofs and empirical results.

**Essential References Not Discussed:**

No, as far as I know, the related works are discussed adequately in this paper.

**Experimental Designs Or Analyses:**

Yes, the experimental designs and analysis are sound and valid.

**Methods And Evaluation Criteria:**

Their methods make sense for the MST problem. Their evaluation criteria are comprehensive, including the approximation factor, runtime, and an upper bound on overlap parameter $\gamma$ given by the overlap with the MST solution over various synthetic datasets and real-world datasets.

**Other Comments Or Suggestions:**

I don't have any further comments.

**Other Strengths And Weaknesses:**

Strengths:
This paper introduces a new framework for finding approximate MST for arbitrary metric space, making a meaningful contribution to scalable MST computation. The connection to learning-augmented algorithms is well-motivated. The experiments across multiple datasets show their method provides nearly-optimal MST (with approximation factor close to 1) with 30 to 400 times speed up compared to the traditional exact algorithms. Furthermore, the proposed framework is particularly relevant for large-scale clustering and similarity-based learning tasks.

Overall, I think the framework and analysis in the paper are quite interesting and the experiments are surprisingly good and show the efficiency and effectiveness of their methods.

**Questions For Authors:**

I only have some open-ended questions:

1. In your experiments, the datasets with cluster structures show smaller upper bound on the overlap parameter $\gamma$. In the approximation analysis, would it be possible to have some explicit parameters other than $\gamma$, like a separation parameter, to improve the approximation factor?

2. It is nice that the analysis even does not rely on any balanceness of the forest partition. Would it be possible to get a better approximation factor or trade-off depending on the balanceness of the partition?

**Relation To Broader Scientific Literature:**

This paper is related to a broad scientific literature including MST, clustering, graph algorithms, learning augmented algorithms, etc. The MST algorithm is also widely used in many machine learning and graph algorithms. This paper has also a great impact on these research as well.

**Theoretical Claims:**

I checked the correctness of all proofs including Theorem 3.1, Theorems 4.2 and 4.3 and all relevant Lemmas.

---

> ### Author Rebuttal · Authors · 2025-03-29
>
> Thank you for the feedback on our manuscript! We appreciate the open question you posed. In response to those questions:
>
> 1. This is an interesting question. As potential follow-up work, we have indeed wondered whether we could use assumptions or parameters relating to the structure of the initial forest to further improve our guarantees (e.g., parameters or structures that results from specifically using a k-centering approximation algorithm or another method to form the initial forest).
>
>
> 2. This is a very interesting question and we do have at least a partial answer with regard to the effect of balancedness on runtime. The answer is a little nuanced because it depends on whether you consider the method used to form the initial forest. If we focus solely on the MFC problem with a given forest, it turns out that unbalanced clusters lead to faster runtimes, even for optimally solving the MFC problem. Note that the $\Omega(n^2)$ lower bound on runtime we give in Theorem 3.1 is assuming a worst-case scenario of balanced clusters. It is an interesting open direction to explore more concrete results if we assume a certain level of balance or imbalance.
> The answer changes slightly if you also consider the method used to construct the initial forest. In our experiments we computed the MSTs of our components optimally. Because of this, having unbalanced clusters means computing an optimal MST for a large component, which is computationally expensive and ends up outweighing the runtime benefit we get for solving the MFC step of the overall MST pipeline. Conversely, for balanced clusters the runtimes tend to be much better, and we still obtain good approximation guarantees. These tradeoffs are highlighted in Table 3 of our paper, where we include runtimes for the different steps of the full MFC framework (including forming the initial forest). These observations are all based on one approach for forming an initial forest (k-centering + optimally finding MSTs of components). Continuing to explore tradeoffs using this or other approaches for forming the initial forest is a very interesting open future direction.

---

### Official Review · Reviewer_29qN · 2025-03-14

**Overall Recommendation:** 4

**Summary:**

The paper considers the problem of building Spanning tree of a set of points in a metric space. The goal is to have a tree that is a good approximation of the MST without querying all the Theta(n^2) pairwise distances. The approach proposed is to have an algorithm that starts from an initial forest and then merges the trees building an overall spanning tree. The algorithm proposed achieves constant approximation if the initial forest has a good "overlap"with some optimal MST. In this sense the approach falls in the category of learning-augmented algorithm, i.e., if by some predicting strategy a good initial partial solution is provided, then the remaining steps have good performance guaratee---not achievable in general because of well known hardness results implying Omega(n^2) queries for any constant approximation.

## Update after rebuttal
I kept my positive score. The rebuttal  didn't need to bring any further information.

**Claims And Evidence:**

The theoretical part appear to be sound.

**Essential References Not Discussed:**

N/A

**Experimental Designs Or Analyses:**

See above

**Methods And Evaluation Criteria:**

The experimental analysis of the practical efficacy of the approach is well structured. The authors design and uses different sets of tests with both synthetic (uniformly sampled) and data from real databases. The experiments address 4 questions (runtime, the initial overlap achievable in practice, theoretical bounds vs performance in practice, effect of different datasets and metrics on performance) and the results are convincing.

**Other Comments Or Suggestions:**

-

**Other Strengths And Weaknesses:**

Strenghts:
Theoretical guarantees conditional to the goodness of the initial forest.
A significant set of experimental analyses addressing the individual parts of the approach and how internal (initial setting) and external (dataset structure) affect the performance.

**Questions For Authors:**

-

**Relation To Broader Scientific Literature:**

The approximation guarantee in the framework of general metric speces is to my knowledge new.

**Theoretical Claims:**

I checked some of the lemmas' and theorems' proofs also in the appendix.

---

> ### Author Rebuttal · Authors · 2025-03-29
>
> Thank you for the review and helpful feedback on our manuscript!

---

### Official Review · Reviewer_udd8 · 2025-03-15

**Overall Recommendation:** 4

**Summary:**

This paper considers the problem of finding subquadratic algorithms for metric MST (minimum spanning tree in a metric space) by leveraging learning-augmented algorithms. The goal is to break through strong lower bounds on runtime: the authors motivate the present work with the result that in an arbitrary metric space, it is impossible to achieve *any* constant factor approximation without knowing $\Omega(n^2)$ edges. In practice, such algorithms are not scalable. To this end, they assume an algorithm has access to an initial forest (on the whole vertex set) that can be thought of as an approximation to a spanning tree (e.g., the output of the first few iterations of Kruskal's algorithm). Given this, the goal is to find an algorithm that solves the Metric Forest Completion problem (MFC), which is to find a minimum-weight set of edges to add to the forest to produce a full spanning tree. This is a connectivity augmentation problem that, to solve to optimality, again requires $\Omega(n^2)$ queries. However, the approach taken here is to instead look for an approximation algorithm that takes subquadratic time. It is important to note that the objective used for computing the approximation ratio is the weight of all edges in the final tree (that of the initial forest and the augmenting set); the authors note that, without counting the initial forest, one cannot achieve good approximations.

There are three main contributions:

- For initial forests with $o(n)$ components, a 2.62 approximation algorithm for MFC with query complexity $o(n^2)$.
- A notion of error (coined $\gamma$-overlap) between the initial forest and the optimal MST, and an analysis that the above algorithm's approximation for the MST problem (not the MFC problem) can be parameterized by $\gamma$: specifically, it achieves a $(2\gamma + 1)$. In particular, for perfect overlap ($\gamma = 1$, meaning all edges in the initial forest are in an optimal MST), this algorithm is a 3-approximation for MST.
- Experiments on real and synthetic datasets (with Euclidean and non-Euclidean metrics) showing that fast heuristics like the $k$-center algorithm can be used to generate an initial forest, and, for datasets with underlying structure, with small $\gamma$. They also show that the approximation ratio tends to be notably better than the theoretical bound.


## update after rebuttal:
 The authors answered the questions well and I maintain my evaluation.

**Claims And Evidence:**

Yes, theoretical claims are supported with proofs (deferred to Appendix), and experimental claims are supported with a detailed explanation of dataset construction and analysis.

**Essential References Not Discussed:**

I did not notice any important related works that are missing.

**Experimental Designs Or Analyses:**

I read through the experimental design and analyses included the appendix, and found the methods completely reasonable.

**Methods And Evaluation Criteria:**

Yes, in addition to proof-of-concept synthetic datasets, the authors use a number of real-world datasets that are used as benchmarks in the clustering literature, and standard measures of distance on that data.

**Other Comments Or Suggestions:**

While the paper is very well-written and has very nice figures, I think the related work in the appendix on MSTs with predictions is important context with which to situate the present contribution, and therefore should be promoted to the main body.

You may also consider including additional references of results on using the predictions framework to improve runtime.

**Other Strengths And Weaknesses:**

There are several interesting contributions here. One is the metric forest completion problem (MFC), which is a model admitting an informative analysis even when $\lambda$ is potentially large (the only assumption is that the initial forest has a sufficiently small number of components). Another is the learning-augmented framework. While previous papers have considered MSTs with predictions, the present framework is distinct, as it is practically and conceptually of interest to obtain subquadratic algorithms for this problem. What is further interesting and distinct here is that the warm-start initial forest is not necessarily some black-box, machined-learned prediction (although it could be), but can be generated from the data itself (e.g., by performing the $k$-center heuristic), and indeed the authors show that doing so is effective in practice. Finally, the authors devise a novel error metric ($\gamma$-overlap) for the "distance" between a spanning forest and an optimal MST that is amenable to analysis; this is interesting as there seemingly isn't a natural / obvious one here.

**Questions For Authors:**

Did you consider comparing the performance of your algorithm against heuristics that compute a full MST in subquadratic time (e.g., if there are any common ones used in practice to overcome the runtime bottleneck)?

Are there any analogues of your MFC problem for other combinatorial problems? Particularly, the idea of computing the approximation ratio with respect to the total cost rather than just the cost of the new edges $M$ (which as you point out is necessary in order to obtain meaningful results)?

**Relation To Broader Scientific Literature:**

This paper contributes to the recently popular framework of algorithms with predictions -- a type of beyond-worst-case approach to algorithm design that seeks to improve algorithmic performance by augmenting algorithms with a (often machine-learned) "prediction" of some parameter of the input (in this case, the initial forest is a "prediction" of the MST). A number of prior works have considered, as the authors do here, this framework for improving *runtime* for popular algorithmic problems (e.g., binary search, min-cost perfect matching, maximum flow). The MST problem has also been considered in the predictions framework, but with different models for what information is learning-augmented, specifically, they all consider predictions on edge weight. The authors here are also distinct in their goal of obtaining subquadratic  algorithms.

**Theoretical Claims:**

I did not check proofs, which are all deferred to the appendix.

---

> ### Author Rebuttal · Authors · 2025-03-29
>
> Thank you for your feedback on our manuscript!
>
> Regarding the section in the appendix on MSTs with predictions, we agree that this related work provides good context for our work and would be nice to include in the main text. Ultimately we moved this to the appendix because of space constraints. However, if the manuscript is accepted this will allow 1 additional page of space, which we would use to elaborate on related work in the main text.
>
> We agree that comparing our approach against other heuristics would be interesting, but the main issue is that only existing heuristics that we are aware of apply only to Euclidean metrics. Our manuscript focuses on algorithms that work for arbitrary metric spaces, and hence we felt that the best experiment section for this paper would provide a concrete comparison between methods that come with theoretical guarantees for arbitrary metric spaces, which in this case amounts to comparing our new approach against the naive quadratic approach.
>
> Finally, regarding your question about analogues of the MFC problem for other combinatorial problems—at present we are not aware of any, but we think this is a very interesting direction to consider in future work. Thanks for the suggestion!

---

### Decision · Program_Chairs · 2025-05-01

**Decision:**

Accept (poster)

**Comment:**

All reviewers like the results of this submission and are convinced that approximating a minimum spanning tree of a point set by a learning-augmented algorithm in subquadratic time is an important result, which has applications for various problems (e.g. hierarchical clustering). Also the technical contribution is interesting and non-trivial and in the experiments the proposed method performed even better than in the worst case. Overall, I consider this to be a strong contribution to ICML.